# FairDD: Fair Dataset Distillation via Synchronized Matching

## Abstract

Condensing large datasets into smaller synthetic counterparts has demonstrated its promise for image classification. However, previous research has overlooked a crucial concern in image recognition: ensuring that models trained on condensed datasets are unbiased towards protected attributes (PA), such as gender and race. Our investigation reveals that dataset distillation (DD) fails to alleviate the unfairness towards minority groups within original datasets. Moreover, this bias typically worsens in the condensed datasets due to their smaller size. To bridge the research gap, we propose a novel fair dataset distillation (FDD) framework, namely FairDD, which can be seamlessly applied to diverse matching-based DD approaches, requiring no modifications to their original architectures. The key innovation of FairDD lies in synchronously matching synthetic datasets to PA-wise groups of original datasets, rather than indiscriminate alignment to the whole distributions in vanilla DDs, dominated by majority groups. This synchronized matching allows synthetic datasets to avoid collapsing into majority groups and bootstrap their balanced generation to all PA groups. Consequently, FairDD could effectively regularize vanilla DDs to favor biased generation toward minority groups while maintaining the accuracy of target attributes. Theoretical analyses and extensive experimental evaluations demonstrate that FairDD significantly improves fairness compared to vanilla DD methods, without sacrificing classification accuracy. Its consistent superiority across diverse DDs, spanning Distribution and Gradient Matching, establishes it as a versatile FDD approach.

## 1 Introduction

Deep learning has witnessed remarkable success in computer vision, particularly with recent breakthroughs in vision models (Oquab et al., 2023; Kirillov et al., 2023; Radford et al., 2021; Li et al., 2022; Zhou et al., 2023). Their vision backbones, such as ResNet (He et al., 2015) and ViT (Dosovitskiy et al., 2020), are data-hungry models that require extensive amounts of data for optimization. Dataset Distillation (DD) (Wang et al., 2018; Zhao & Bilen, 2021; 2023; Cazenavette et al., 2022; Wang et al., 2022; Lee et al., 2022b; Cui et al., 2022; Loo et al., 2023; Guo et al., 2023; He & Zhou, 2024; Zhao et al., 2024) provides a promising solution to alleviate this data requirement by condensing the original large dataset into more informative and smaller counterparts (Mehrabi et al., 2021; Chung et al., 2023). Despite its appeal, existing DD researches focus on ensuring that models trained on condensed datasets perform comparable accuracy to those trained on the original dataset in terms of target attributes (TA) (Cui et al., 2024; Lu et al., 2024; Vahidian et al., 2024). However, they have overlooked enabling the fairness of trained models with respect to protected attributes (PA).

Unfairness typically arises from imbalanced sample distributions among PA in the empirical training datasets. When the original datasets suffer from the PA imbalance, the corresponding datasets condensed by vanilla DDs inherit and amplify this bias in Fig. 1(a). Since vanilla DDs tend to cover TA distribution for image classification, and as a result, it naturally leads to more synthetic samples located in majority groups compared to minority groups w.r.t. PA, as shown in Fig. 1(b). In this case, these condensed datasets retain the imbalance between protected attributes, thereby rendering the model trained on them unfair. Moreover, the reduced size of the condensed datasets typically amplifies the bias present in the original datasets, especially when there is a significant gap in size between the original and condensed datasets, such as image per class (IPC) 1000 vs. 10. Therefore, it is worthwhile to broaden the scope of DD to encompass both TA accuracy and PA fairness. Recent

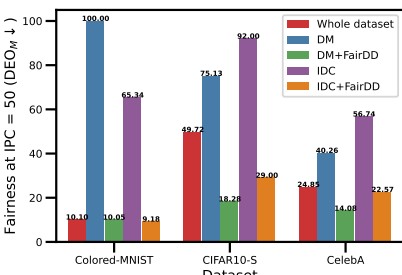

(a) Fairness comparison on original and distilled datasets using different DDs.

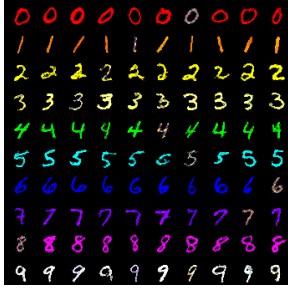

(b) Condensed datasets on C-MNIST (FG) using DM.

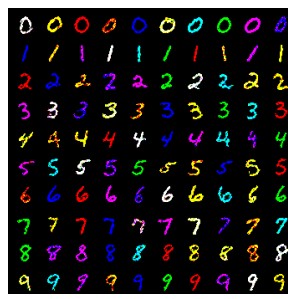

(c) Condensed datasets using FairDD+DM.

Figure 1: Condensed datasets fail to mitigate and amplify the bias present in original datasets. (a) depicts the bias against minority groups of the model trained on condensed datasets, showing that vanilla DDs exacerbate the unfairness (IPC=50). However, incorporating FairDD into vanilla DDs could effectively mitigate these biases. (b) and (c) distill C-MNIST (FG) (IPC=10), where each class contains 10 colors and individual colors dominate different categories as PA. (b) shows that each class (row) is biased to the majority color in the condensed dataset via vanilla DDs. (c) preserves PA of minority groups and mitigates color imbalance in the condensed dataset via DM+FairDD.

works attempt to address the class/TA-level long-tailed phenomenon Zhao et al. (2024) and spurious correlations (Cui et al., 2024) to improve the classification performance (Vahidian et al., 2024; Wang et al., 2024), but the exploration on Fair Dataset Distillation (FDD) is still blank.

To the best of our knowledge, we are the first to propose a novel FDD framework, FairDD, which enables PA fairness in the model trained on the condensed datasets, regardless of PA imbalance in the original data. However, simultaneously maintaining TA accuracy and improving PA fairness is challenging, as the algorithm must properly balance emphasis across all groups, i.e., majority groups primarily for TA distributional coverage and minority groups primarily for PA bias mitigation.

FairDD tackles this challenge by **(1)** partitioning the empirical training distribution into different groups according to PA and decomposing the single alignment target of vanilla DDs into PA-wise sub-targets. **(2)** synchronously matching synthetic samples to these PA groups, which equally bootstraps synthetic datasets to each PA group without involving the specific group size. In doing so, we reformulated the optimization objectives of vanilla DDs into fairDD-style versions. This allows FairDD to mitigate the effect of the imbalanced PA to the generation of $\mathcal{S}$ and to prevent $\mathcal{S}$ from collapsing into the majority group. Meanwhile, FairDD could also achieve the comprehensive coverage of the entire distribution for TA accuracy as shown in Fig. 1(c). In addition, FairDD requires no modification to existing DDs or additional modules. We provide a theoretical guarantee that FairDD could improve PA fairness while maintaining TA accuracy.

Extensive experiments demonstrate that our proposed framework effectively mitigates the unfairness in datasets of highly diverse bias. FairDD substantially improves model fairness on condensed datasets compared to various vanilla DDs. FairDD demonstrates its versatility across diverse DD paradigms, including Distribution Matching (DM) and Gradient Matching (GM). Note that we do not apply FairDD to Trajectory Matching (TM) as doing so would require extra model trajectories trained on minority groups, in which the corresponding models would suffer from overfitting due to their limited samples. Our paper makes the following main contributions:

- To the best of our knowledge, our research is the first attempt to incorporate fairness issues into DD explicitly. We reveal that vanilla DDs fail to mitigate the bias in original datasets and may exacerbate it due to the limited synthetic samples, leading to severe PA bias in the model trained by the resulting condensed dataset.

- We introduce a novel FDD framework called FairDD, which proposes synchronized matching to align synthetic samples to all PA groups partitioned from the original data distribution. This allows the generated synthetic samples to be agnostic to PA imbalance of original datasets and maintain the overall distributional coverage of TA.

- FairDD requires no alternations to the original architectures of vanilla DDs, in which it only needs to revise the vanilla optimization objectives as a group-level formulation. We also provide theoretical analyses to guarantee the fairness and accuracy of synthetic samples.

- Extensive empirical experiments demonstrate that FairDD mitigates the unfairness of vanilla DDs by a large margin. The consistent superiority across diverse DDs, including DM and GM, illustrates that FairDD is a generalist for FDD.

## 2 PRELIMINARIES

**Dataset Distillation.** Given a vast dataset $\mathcal{T} = \{(x_i, y_i)\}_{i=1}^{N}$, the goal of DD is to condense the original dataset $\mathcal{T}$ into a smaller dataset $\mathcal{S} = \{(x_i, y_i)\}_{i=1}^{M}$ via distillation algorithm Alg with a nerual network, parameterized by $\theta$. Randomly initialized classification network $g_\psi$ should maintain the same empirical risk whether it is trained on $\mathcal{S}$ or $\mathcal{T}$.

$$\mathcal{S}^* = \arg\min_{S} \text{Alg}(\mathcal{S}, \mathcal{T}, \theta), \tag{1}$$

$$\mathbb{E}_{\psi \sim \Psi}[\ell(g_\psi; \mathcal{S})] \simeq \mathbb{E}_{\psi \sim \Psi}[\ell(g_\psi; \mathcal{T})], \tag{2}$$

where $\Psi$ and $\ell(\cdot)$ represent the parameter space and loss function, respectively. The pioneering work (Wang et al., 2018) formulates Alg as a bi-level optimization problem. However, such an optimization process is time-consuming and unsTable Recent works circumvent it and propose surrogate matching objectives to achieve comparable and even better performance. This research line is collectively referred to as the DMF, and our paper primarily studies one-stage GM (Zhao et al., 2020; Zhao & Bilen, 2021) and DM (Zhao & Bilen, 2023; Wang et al., 2022; Zhao & Bilen, 2022). We leave two-stage trajectory-matching (TM) for future exploration.

**Visual Fairness** Visual fairness is an important field to mitigate discrimination against minority groups. Group fairness requires no statistical disparity to different groups in terms of PA, such as race and gender. This means that an ideal fair model should make independent predictions between TA and PA. One of the common fairness criteria is equalized odds (EO), which computes the prediction accuracy of PA conditioned on TA, to evaluate the level of conditional independence between PA and TA. We use two types of difference of equalized odds $\text{DEO}_M$ and $\text{DEO}_A$ from the worst and averaged levels. Formally, given the PA set $\mathcal{A} = \{a_1, a_2, ..., a_p\}$, $\text{DEO}_M$ and $\text{DEO}_A$ (Jung et al., 2021) can be formulated mathematically as follows:

$$\text{DEO}_M = \max_{y \in \mathcal{Y}} \max_{a_i, a_j \in \mathcal{A} \& a_i \neq a_j} \left| P(\hat{Y} = y | Y = y, A = a_i) - P(\hat{Y} = y | Y = y, A = a_j) \right|,$$

$$\text{DEO}_A = \max_{y \in \mathcal{Y}} \max_{a_i, a_j \in \mathcal{A} \& a_i \neq a_j} \left| P(\hat{Y} = y | Y = y, A = a_i) - P(\hat{Y} = y | Y = y, A = a_j) \right|.$$

## 3 A CLOSE LOOK AT DATASET DISTILLATION FROM FAIRNESS

**A unified perspective for Data Match Framework.** The essence of the DMF lies in choosing the target signs of original samples that effectively represent their characteristics for image recognition, and then aligning these signals as a proxy task to optimize the condensed dataset. The target signal $\phi(x; \theta)$ is typically the key information from feature extraction or optimization process using a randomly initialized network parameterized by $\theta$. For example, GM aligns the gradient information produced by $\mathcal{T}$ with that of the condensed $\mathcal{S}$. Instead, DM matches the embedding distributions of $\mathcal{T}$ and $\mathcal{S}$. As for these approaches in DMF. we can unify the optimization objective as $\mathcal{L}(\mathcal{S}; \theta, \mathcal{T})$:

$$\mathcal{L}(\mathcal{S}; \theta, \mathcal{T}) := \sum_{y \in \mathcal{Y}} \mathcal{D}\big(\mathbb{E}[\phi_{x \sim \mathcal{T}_y}(x; \theta)], \mathbb{E}[\phi_{x \sim \mathcal{S}_y}(x; \theta)]\big), \tag{3}$$

where $\mathbb{E}[\phi_{x \sim \mathcal{T}_y}(x; \theta)] \in \mathbb{R}^C$ and $\mathbb{E}[\phi_{x \sim \mathcal{S}_y}(x; \theta)] \in \mathbb{R}^C$ are represented expectation vectors of the target signs on $\mathcal{T}$ and $\mathcal{S}$, respectively. $\mathcal{D}(\cdot, \cdot)$ is a distance function. In DMF, MSE is adopted in DM and DREAM, and MAE is used in IDC. Also, cosine distance is involved in DC.

**Why does vanilla DD fail to mitigate PA imbalance?** Given the PA set $\mathcal{A}$ in $\mathcal{T}$, let us define the class-level sample ratio $\mathcal{R}_y = \{r_y^{a_1}, r_y^{a_2}, ..., r_y^{a_p}\}$, where $r_y^{a_i} = |\mathcal{T}_y^{a_i}|/|\mathcal{T}_y|$, and $|\cdot|$ represents the cardinal number of a set. Current DD paradigms focus on preserving TA representativeness for image recognition. Here, we decompose the whole expectation into the expectation of PA-wise groups, i.e, $\mathbb{E}[\phi_{x \sim \mathcal{T}_y}(x; \theta)] = \sum_{a_i \in \mathcal{A}} r_y^{a_i} \mathbb{E}[\phi_{x \sim \mathcal{T}_y^{a_i}}(x; \theta)]$, and thus Eq. 3 can be rewritten as follows:

$$\mathcal{L}(\mathcal{S}; \theta, \mathcal{T}) := \sum_{y \in \mathcal{Y}} \mathcal{D}(\sum_{a_i \in \mathcal{A}} r_y^{a_i} \mathbb{E}[\phi_{x \sim \mathcal{T}_y^{a_i}}(x; \theta)], \mathbb{E}[\phi_{x \sim \mathcal{S}_y}(x; \theta)]). \tag{4}$$

From Eq. 4, the optimization objective of class $y$ is weighted by the sample ratio $r_y^{a_i}$ from different groups. When $\mathcal{T}$ suffers from PA imbalance, e.g., $r_y^{a_j} \gg \sum_{i \neq j} r_y^{a_i}$, the majority group indexed by $i$ contributes more to the alignment compared to minority groups, as present in Fig. 2(a). In

other words, $\mathcal{S}$ tends to produce more samples belonging to group $i$ for the total loss minimization. Therefore, the objective of vanilla DDs suffers from the PA imbalance within $\mathcal{T}$.

Next, we further study how the resulting $\mathcal{S}$ is affected by sample ratio $r_y^{a_i}$ of different groups. To this end, we assume that the optimization process could reach the optimal solution for each class, and as a result, the final resulting $\mathcal{S}$ satisfies the condition that the derivative of the objective function with respect to $\mathbb{E}[\phi_{x \sim \mathcal{S}_y}(x; \theta)]$ equals 0. Formally. we have the following mathematical equation:

$$\frac{\partial \mathcal{L}(\mathcal{S}_y; \theta, \mathcal{T}_y)}{\partial \mathbb{E}[\phi_{x \sim \mathcal{S}_y}(x; \theta)]} = 0 \implies \frac{\partial \mathcal{D}(\sum_{a_i \in \mathcal{A}} r_y^{a_i} \mathbb{E}[\phi_{x \sim \mathcal{T}_y^{a_i}}(x; \theta)], \mathbb{E}[\phi_{x \sim \mathcal{S}_y}(x; \theta)])}{\partial \mathbb{E}[\phi_{x \sim \mathcal{S}_y}(x; \theta)]} = 0 \tag{5}$$

Now, let's delve into the specific distance metrics used in vanilla DDs, where the most commonly used metrics are MAE, MSE, and cosine distance. We respectively analyze that the optimal point of $\mathbb{E}[\phi_{x \sim \mathcal{S}_y}(x; \theta)]$ could reach under each of these metrics.

$$\frac{\partial \mathcal{L}(\mathcal{S}_y; \theta, \mathcal{T}_y)}{\partial \mathbb{E}[\phi_{x \sim \mathcal{S}_y}(x; \theta)]} = 0 \implies \mathbb{E}[\phi_{x \sim \mathcal{S}_y}(x; \theta)] = \begin{cases} \sum_{a_i \in \mathcal{A}} r_y^{a_i} \mathbb{E}[\phi_{x \sim \mathcal{T}_y^{a_i}}(x; \theta)], & \text{For MAE} \\ \sum_{a_i \in \mathcal{A}} r_y^{a_i} \mathbb{E}[\phi_{x \sim \mathcal{T}_y^{a_i}}(x; \theta)], & \text{For MSE} \\ \lambda \sum_{a_i \in \mathcal{A}} r_y^{a_i} \mathbb{E}[\phi_{x \sim \mathcal{T}_y^{a_i}}(x; \theta)], & \text{For cosine distance} \end{cases} \tag{6}$$

Where $\lambda$ is a scalar, equaling $\frac{\|\mathbb{E}[\phi_{x \sim \mathcal{S}_y}(x; \theta)]\|_2}{\|\sum_{a_i \in \mathcal{A}} r_y^{a_i} \mathbb{E}[\phi_{x \sim \mathcal{T}_y^{a_i}}(x; \theta)]\|_2}$. Eq. 6 presents that the expectation of synthetic samples $\mathbb{E}[\phi_{x \sim \mathcal{S}_y}(x; \theta)]$ ultimately converges to an average on expectations of all PA groups, weighted by their respective sample ratios $r_y^{a_i}$. This indicates that vanilla DDs naturally favor majority groups, causing $\mathcal{S}$ to shift towards them and inherit their biases.

Therefore, when original datasets suffer from PA imbalance, e.g., $r_y^{a_j} \gg \sum_{i \neq j} r_y^{a_i}$, the unfairness of the synthetic dataset stems from two different aspects: 1) **The majority group renders synthetic samples to locate its region from Eq. 6.** 2) According to Eq. 4, the large sample quantities of the majority group contribute more to the total loss. As a result, **minority groups experience higher loss during testing, which limits the model to represent them accurately.** These factors prompt us to reduce the impact of PA imbalance on the generation of $\mathcal{S}$.

## 4 FairDD

### 4.1 Overview

In this paper, we propose a novel FDD framework that achieves both PA fairness and TA accuracy for the model trained on its generation $\mathcal{S}$, regardless of whether the original datasets exhibit PA fairness. As illustrated in Fig. 2(b), FairDD first partitions the dataset into different groups w.r.t. PA and then introduces an effective synchronized matching to equally align $\mathcal{S}$ with each group within $\mathcal{T}$. Compared to vanilla DDs, which simply pull the synthetic dataset toward the whole dataset center that is biased toward the majority group in the synthetic dataset, FairDD proposes a group-level synchronized alignment, in which each group attracts the synthetic data toward itself, thus forcing it to move farther from other groups. This "pull-and-push" process prevents the generation from collapsing into majority groups (fairness) and ensures class-level distributional coverage (accuracy).

### 4.2 Methods

As mentioned in Sec. 3, vanilla DDs fail to mitigate PA imbalance and even amplify the discrimination. The relation behind the failure is that the majority group dominates the generation direction of $\mathcal{S}$ and leads to the resulting $\mathcal{S}$ inheriting the PA imbalance, i.e., preference to fitting to the majority group. To avoid the synthetic samples collapsing into the majority group, we decompose the single target (dominated by the majority group) into PA-wise sub-targets, and simultaneously align $\mathcal{S}$ with these sub-targets, without incorporating the specific sample ratio of each group into the optimization objective. In this way, we obtain the unified objective function of FairDD:

$$\mathcal{L}_{FairDD}(\mathcal{S}; \theta, \mathcal{T}) := \sum_{y \in \mathcal{Y}} \sum_{a_i \in \mathcal{A}} \mathcal{D}(\mathbb{E}[\phi_{x \sim \mathcal{T}_y^{a_i}}(x; \theta)], \mathbb{E}[\phi_{x \sim \mathcal{S}_y}(x; \theta)]). \tag{7}$$

The reformulation forms synchronized matching, where different sub-targets attempt to pull $\mathcal{S}$ into their corresponding PA regions. Each PA group holds equal importance in generating $\mathcal{S}$, ultimately converging to a balanced (fair) status. Subsequently, we present a theoretical analysis illustrating how FairDD effectively mitigates PA imbalance and guarantees coverage across the entire TA distribution.

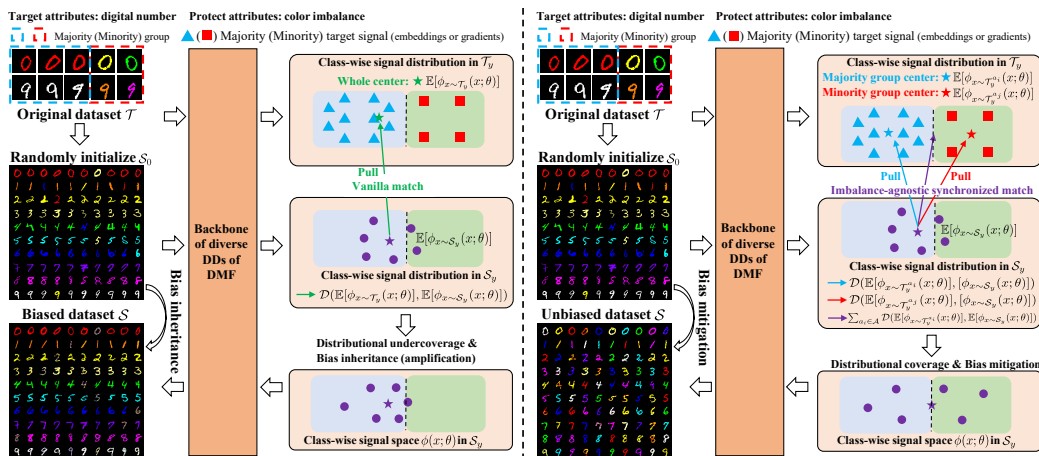

(a) The pipeline of vanilla DDs.          (b) The pipeline of FairDD.

Figure 2: Comparison between (a) vanilla DDs and (b) FairDD. Taking C-MNIST (FG) for example (IPC = 10), vanilla DDs directly align $\mathcal{S}$ (random initialization) with the whole distribution regardless of majority or minority groups. This causes $\mathcal{S}$ to suffer from distributional coverage and inherit (even amplify) the bias of $\mathcal{T}$. Instead, FairDD first groups target signals of $\mathcal{T}$ and then proposes to align $\mathcal{S}$ (random initialization) with respective group centers. With this synchronized matching, $\mathcal{S}$ is simultaneously pulled by all group centers in a batch. This prevents the condensed dataset $\mathcal{S}$ from being biased towards the majority group, allowing it to better cover the distribution of the original dataset $\mathcal{T}$. We can observe that each class in the resulting $\mathcal{S}$ incorporates multiple colors from the minority color groups and mitigates the bias of $\mathcal{T}$, originally dominated by majority colors.

**Theorem 4.1.** *For any PA set $\mathcal{A}$, network parameters $\theta$, and target signs $\phi(\cdot)$, $\mathcal{L}_{FairDD}(\mathcal{S};\theta,\mathcal{T})$ could mitigate the influence of PA imbalance of original datasets on generating synthetic samples. Especially when $\mathcal{D}(\cdot)$ is MAE or MSE, synchronized matching ensures that the signal expectation of $\mathcal{S}$ is situated at the center of the expectation across all PA groups within $\mathcal{T}$*

*Proof.* We assume that $\mathbb{E}[\phi_{x \sim \mathcal{S}_y}(x;\theta)]$ could reach the optimal solution for each class. Consequently, we have $\partial \mathcal{L}_{FairDD}(\mathcal{S}_y;\theta,\mathcal{T}_y)/\partial \mathbb{E}[\phi_{x \sim \mathcal{S}_y}(x;\theta)] = 0$, and derive the respective optimal solution:

$$\frac{\partial \mathcal{L}_{FairDD}(\mathcal{S}_y;\theta,\mathcal{T}_y)}{\partial \mathbb{E}[\phi_{x \sim \mathcal{S}_y}(x;\theta)]} = 0 \implies \mathbb{E}[\phi_{x \sim \mathcal{S}_y}(x;\theta)] = \begin{cases} \frac{1}{|\mathcal{A}|} \sum_{a_i \in \mathcal{A}} \mathbb{E}[\phi_{x \sim \mathcal{T}_y^{a_i}}(x;\theta)], & \text{For MAE} \\ \frac{1}{|\mathcal{A}|} \sum_{a_i \in \mathcal{A}} \mathbb{E}[\phi_{x \sim \mathcal{T}_y^{a_i}}(x;\theta)], & \text{For MSE} \\ \frac{\lambda}{|\mathcal{A}|} \sum_{a_i \in \mathcal{A}} \mathbb{E}[\phi_{x \sim \mathcal{T}_y^{a_i}}(x;\theta)], & \text{For cosine distance} \end{cases} \quad (8)$$

According to Eq. 8, the resulting $\mathbb{E}[\phi_{x \sim \mathcal{S}_y}(x;\theta)]$ are independent on the sample ratio $\mathcal{R}_y$, indicating the corresponding $S$ unaffected by $\mathcal{R}_y$. As a result, the condensed $\mathcal{S}$ will not be dominated by majority groups that happened in vanilla DDs. All PA centers contribute equally to the generation of $\mathcal{S}$, which succeeds in mitigating the PA imbalance of $\mathcal{T}$. Especially when $\mathcal{D}(\cdot)$ is MAE or MSE, the expectation of target signs of $\mathcal{S}$ is equal to the arithmetic mean of centers of all PA groups. This shows that $\mathcal{S}$ generated by FairDD is not biased towards any groups.

Although we mitigate the bias inheritance in vanilla DDs by synchronously aligning $\mathcal{S}$ to fine-grained PA-wise groups, it is also crucial to investigate whether $\mathcal{L}_{FairDD}(\mathcal{S};\theta,\mathcal{T})$ (synchronized matching) ensures that the resulting $\mathcal{S}$ achieves comprehensive distributional coverage for $\mathcal{T}$. As mentioned above, $\mathcal{L}(\mathcal{S};\theta,\mathcal{T})$ matches $\mathcal{S}$ and $\mathcal{T}$ in a global view to fully cover $\mathcal{T}$'s distribution. Below, we provide a theoretical guarantee that $\mathcal{L}_{FairDD}(\mathcal{S};\theta,\mathcal{T})$ could provide the same or even more comprehensive coverage compared to $\mathcal{L}(\mathcal{S};\theta,\mathcal{T})$ when $\mathcal{D}(\cdot,\cdot)$ is a convex distance function, which is commonly used in diverse DDs [1].

---

[1] Emperical experiments show FairDD also can cover the TA distributions when $\mathcal{D}(\cdot,\cdot)$ is not convex.

**Theorem 4.2.** *For any PA set $\mathcal{A}$ and target signs $\phi_\theta(\cdot)$, $\mathcal{L}_{FairDD}(\mathcal{S}; \theta, \mathcal{T})$ is the upper bound of vanilla unified objective $\mathcal{L}(\mathcal{S}; \theta, \mathcal{T})$, i.e., $\mathcal{L}_{FairDD}(\mathcal{S}; \theta, \mathcal{T}) \geq \mathcal{L}(\mathcal{S}; \theta, \mathcal{T})$, when $\mathcal{D}(\cdot, \cdot)$ is convex. Optimizing $\mathcal{L}_{FairDD}(\mathcal{S}; \theta, \mathcal{T})$ can guarantee the comprehensive distribution coverage for $\mathcal{T}$.*

*Proof.*

$$\mathcal{L}(\mathcal{S}; \theta, \mathcal{T}) = \sum_{y \in \mathcal{Y}} \mathcal{D}\big(\mathbb{E}[\phi_{x \sim \mathcal{T}_y}(x; \theta)], \mathbb{E}[\phi_{x \sim \mathcal{S}_y}(x; \theta)]\big)$$

$$= \sum_{y \in \mathcal{Y}} \mathcal{D}\big(\sum_{a_i \in \mathcal{A}} r_y^{a_i} \mathbb{E}[\phi_{x \sim \mathcal{T}_y^{a_i}}(x; \theta)], \mathbb{E}[\phi_{x \sim \mathcal{S}_y}(x; \theta)]\big)$$

$$\leq \sum_{y \in \mathcal{Y}} \sum_{a_i \in \mathcal{A}} r_y^{a_i} \mathcal{D}\big(\mathbb{E}[\phi_{x \sim \mathcal{T}_y^{a_i}}(x; \theta)], \mathbb{E}[\phi_{x \sim \mathcal{S}_y}(x; \theta)]\big) \qquad (9)$$

$$\leq \sum_{y \in \mathcal{Y}} \sum_{a_i \in \mathcal{A}} \mathcal{D}\big(\mathbb{E}[\phi_{x \sim \mathcal{T}_y^{a_i}}(x; \theta)], \mathbb{E}[\phi_{x \sim \mathcal{S}_y}(x; \theta)]\big) \qquad (10)$$

$$= \mathcal{L}_{FairDD}(\mathcal{S}; \theta, \mathcal{T})$$

Eq. 9 is obtained according to the Jensen Inequality, and Eq. 10 is given since group ratios are smaller than one. $\mathcal{L}_{FairDD}(\mathcal{S}; \theta, \mathcal{T})$ serves as the upper bound of $\mathcal{L}(\mathcal{S}; \theta, \mathcal{T})$, meaning that minimizing $\mathcal{L}_{FairDD}(\mathcal{S}; \theta, \mathcal{T})$ ensures the minimization of $\mathcal{L}(\mathcal{S}; \theta, \mathcal{T})$. Hence, optimizing $\mathcal{S}$ with FairDD can guarantee the distributional coverage, at least as comprehensive as $\mathcal{L}(\mathcal{S}; \theta, \mathcal{T})$ tailored for accuracy.

## 5 EXPERIMENT

### 5.1 EXPERIMENT SETUP

**Datasets** Comprehensive experiments have been conducted on publicly available datasets of diverse biases, including foreground bias (FG), background bias (BG), BG & FG bias, and real-world bias. C-MNIST (FG) is a variant of MNIST (LeCun et al., 2010) used to evaluate model fairness, where the handwriting numbers in each class are painted with ten different colors. To correlate the TA (digital number) and PA (color) within the training dataset, each training class is predominantly associated with one color according to the same biased ratio (BR), while the remaining samples are evenly painted with the other nine colors. BR is the ratio of the majority group samples to the total samples across all groups. For the test dataset, we evenly paint the numbers for each class with ten colors to test the model bias trained on $\mathcal{S}$. C-MNIST (BG) adopts the same operation on the background and keeps the foreground unchanged. Colored-FMNIST (FG) is the modified version of Fashion-MNIST, originally aiming to classify object semantics. Like C-MNIST (FG), we color the objects for the training and test datasets. Colored-FMNIST (BG) paints the background similarly to C-MNIST (BG). CIFAR10-S (BG & FG) introduces a PA by applying grayscale or not to CIFAR10 samples. Following Wang et al. (2020), we grayscale a portion of the training images, correlating TA and PA among different classes. For fairness evaluation, we duplicate the test images, apply grayscale to the copies, and add them to the test dataset. We also test FairDD on the real-world facial dataset CelebA, a widely used fairness dataset. We follow the common practice of treating `attractive` attribute as TA and `gender` as PA (evaluations on other attributes refer to Appendix B).

**Baselines & Evaluation metrics** FairDD is a general fairness framework applicable to diverse DDs in DMF. We apply FairDD to diverse DMF approaches including DM method DM (Zhao & Bilen, 2023) and GM methods DC (Zhao et al., 2020), IDC (DC version) (Kim et al., 2022), and DREAM (DC version) (Liu et al., 2023). To provide an overall evaluation for model bias toward PA, we use $\text{DEO}_M(\downarrow) \in [0, 100]$ and $\text{DEO}_A(\downarrow) \in [0, 100]$ to measure the worst and average fairness levels. Also, we report accuracy($\uparrow$) to assess the model's prediction of TA. We also provide a comparison with MTT in Appendix L. Sometimes, we will abuse DM+FairDD and FairDD for clarification.

**Implementation details** We default to BR of 0.9 for all synthetic original datasets to induce significant PA skew. In Table 24, we conduct the ablation study on BR. All baselines are reproduced using official implementations. FairDD doesn't introduce extra hyperparameters or learnable parameters. Experiments are conducted on PyTorch 2.0.0 with a single NVIDIA RTX 3090 24GB GPU.

### 5.2 MAIN RESULTS

We use distilled datasets $\mathcal{S}$ from different DDs to train and evaluate ConvNet with the same parameters, and then report the corresponding fairness and accuracy. *Random* refers to sampling defined IPC from the original dataset to create smaller datasets. Besides, *Whole* means we train the model using the entire training dataset without distillation or sampling.

Table 1: Fairness comparison on diverse IPCs. The best results are highlighted in **bold**.

| Methods / Dataset | IPC | Random | | DM | | DM+FairDD | | DC | | DC+FairDD | | IDC | | IDC+FairDD | | DREAM | | +FairDD | | Whole | |
|---|---|---|---|---|---|---|---|---|---|---|---|---|---|---|---|---|---|---|---|---|---|
| | | DEO$_M$ | DEO$_A$ | DEO$_M$ | DEO$_A$ | DEO$_M$ | DEO$_A$ | DEO$_M$ | DEO$_A$ | DEO$_M$ | DEO$_A$ | DEO$_M$ | DEO$_A$ | DEO$_M$ | DEO$_A$ | DEO$_M$ | DEO$_A$ | DEO$_M$ | DEO$_A$ | DEO$_M$ | DEO$_A$ |
| C-MNIST (FG) | 10 | 100.0 | 98.72 | 100.0 | 99.96 | **17.04** | **7.95** | 99.85 | 65.61 | **26.75** | **11.96** | 100.0 | 91.45 | **12.24** | **6.64** | 98.99 | 78.71 | **11.88** | **7.21** | 10.10 | 5.89 |
| | 50 | 100.0 | 99.58 | 100.0 | 91.68 | **10.05** | **5.46** | 46.99 | 20.55 | **18.42** | **8.86** | 65.34 | 34.91 | **9.18** | **5.94** | 52.03 | 26.63 | **18.37** | **7.50** | | |
| | 100 | 100.0 | 88.64 | 99.36 | 66.38 | **8.17** | **4.86** | 45.27 | 17.45 | **22.32** | **9.49** | 64.36 | 35.82 | **11.88** | **6.21** | 69.30 | 33.30 | **11.88** | **6.88** | | |
| C-MNIST (BG) | 10 | 100.0 | 99.11 | 100.0 | 99.97 | **13.42** | **6.77** | 100.0 | 73.60 | **20.66** | **9.94** | 100.0 | 88.30 | **18.61** | **7.50** | 100.0 | 52.06 | **15.31** | **6.83** | 9.70 | 5.78 |
| | 50 | 100.0 | 99.77 | 100.0 | 97.85 | **8.98** | **5.25** | 60.66 | 26.38 | **20.29** | **9.90** | 93.05 | 42.23 | **19.66** | **8.05** | 64.15 | 23.30 | **20.41** | **9.04** | | |
| | 100 | 100.0 | 89.07 | 100.0 | 52.23 | **6.60** | **4.31** | 62.63 | 20.87 | **32.58** | **10.40** | 63.24 | 27.79 | **12.24** | **6.32** | 44.88 | 22.86 | **16.33** | **7.80** | | |
| C-FMNIST (FG) | 10 | 100.0 | 99.18 | 100.0 | 99.05 | **26.87** | **16.38** | 99.40 | 78.96 | **46.80** | **24.01** | 100.0 | 97.27 | **32.33** | **16.80** | 100.0 | 95.17 | **42.00** | **20.87** | 79.20 | 41.72 |
| | 50 | 100.0 | 94.61 | 100.0 | 96.46 | **24.92** | **13.74** | 99.33 | 67.02 | **46.67** | **21.48** | 100.0 | 81.93 | **40.00** | **17.37** | 97.33 | 70.10 | **47.67** | **22.33** | | |
| | 100 | 100.0 | 94.85 | 100.0 | 85.11 | **23.83** | **12.75** | 99.58 | 66.45 | **56.68** | **23.07** | 100.0 | 79.10 | **48.33** | **17.43** | 97.33 | 70.10 | **74.00** | **40.40** | | |
| C-FMNIST (BG) | 10 | 100.0 | 99.40 | 100.0 | 99.68 | **33.05** | **19.72** | 100.0 | 92.91 | **61.75** | **34.88** | 100.0 | 99.40 | **42.00** | **23.80** | 100.0 | 94.70 | **36.00** | **23.50** | 91.40 | 51.68 |
| | 50 | 100.0 | 98.52 | 100.0 | 99.71 | **24.50** | **14.47** | 100.0 | 75.41 | **44.60** | **25.25** | 100.0 | 95.60 | **78.00** | **34.50** | 100.0 | 88.40 | **34.00** | **23.70** | | |
| | 100 | 100.0 | 96.05 | 100.0 | 93.88 | **21.95** | **13.33** | 99.70 | 73.38 | **52.75** | **23.48** | 100.0 | 90.70 | **77.00** | **36.00** | 100.0 | 83.90 | **40.00** | **23.20** | | |
| CIFAR10-S | 10 | 25.04 | 8.29 | 59.20 | 39.31 | **31.75** | **8.73** | 42.23 | 27.35 | **22.08** | **8.22** | 80.70 | 48.38 | **19.90** | **5.28** | 51.80 | 31.43 | **20.80** | **7.77** | 49.72 | 33.17 |
| | 50 | 57.11 | 28.89 | 75.13 | 55.70 | **18.28** | **7.35** | 71.46 | 45.81 | **34.39** | **11.21** | 92.00 | 60.56 | **29.00** | **9.10** | 56.80 | 36.19 | **14.70** | **6.53** | | |
| | 100 | 66.49 | 43.16 | 73.81 | 55.10 | **14.77** | **5.89** | 68.69 | 48.64 | **32.70** | **11.26** | 92.70 | 60.93 | **62.80** | **25.18** | 82.30 | 48.12 | **12.10** | **6.06** | | |
| CelebA | 10 | 10.48 | 9.20 | 30.01 | 28.85 | **9.37** | **5.71** | 15.48 | 14.16 | **6.64** | **5.29** | 34.85 | 34.48 | **8.36** | **4.49** | 40.75 | 36.70 | **9.20** | **5.36** | 24.85 | 24.16 |
| | 50 | 22.88 | 20.32 | 40.26 | 38.81 | **14.08** | **9.87** | 24.89 | 23.83 | **14.33** | **12.92** | 56.74 | 46.50 | **22.57** | **15.15** | 43.57 | 38.53 | **23.62** | **14.29** | | |
| | 100 | 18.67 | 18.01 | 42.63 | 41.12 | **10.93** | **6.65** | 29.00 | 27.52 | **18.16** | **17.04** | 50.99 | 42.66 | **28.27** | **17.63** | 52.51 | 39.34 | **24.87** | **15.36** | | |

(a) C-MNIST (FG).  (b) C-MNIST (BG). (c) C-FMNIST (FG). (d) C-FMNIST (BG).  (e) CIFAR10-S.

Figure 3: Visualization comparison on $\mathcal{S}$ at IPC = 10 for diverse datasets. FairDD successfully mitigates the bias from original datasets in (a) (foreground digital color), (b) background color, (c) foreground object color, (d) background color, and (e) foreground and background grayscale.

**FairDD significantly improves the fairness of vanilla DD approaches**  We provide comprehensive fairness comparisons across various DD paradigms, including DM and DC. As illustrated in Table 21, vanilla DDs fail to mitigate the bias present in the original datasets and even exacerbate unfairness towards biased groups. In C-MNIST (FG), the distilled datasets from DM suffer from severe unfairness at IPC=10 compared to *Whole*, with DEO$_M$ and DEO$_A$ reaching 100.0 and 99.96 vs. 10.10 and 5.89. In some cases, *Random* presents better fairness than vanilla DDs, particularly when dealing with complex objects like CelebA. This suggests that while vanilla DDs effectively condense information into smaller samples, their inductive bias, which favors the majority group, worsens the fairness to the minority group. However, when FairDD is applied to vanilla DDs, there is a significant improvement in fairness performance, with DEO$_M$ dropping substantially from 100.0 to 17.04, and DEO$_A$ decreasing from 99.96 to 7.95 in C-MNIST (FG). This indicates that FairDD's synchronized matching ensures the equal treatment of each group, effectively mitigating the bias that vanilla DDs exacerbate. FairDD further reduces the bias originally present in the original datasets. For example, DC + FairDD outperforms *Whole* in C-FMNIST (FG) and CIFAR10-S, as well as in the real-world dataset CelebA, achieving the overall improvement on DEO$_M$ and DEO$_A$ metrics. Similar performance gains are also observed in other baselines. These results underscore the effectiveness and versatility of FairDD. We provide the visualization comparison of resulting $\mathcal{S}$ in Fig. 3.

**FairDD maintains the comparable and even higher accuracy than vanilla DD approaches**  A fairness framework must maintain TA accuracy in addition to improving fairness across PA groups. We report the TA accuracy of FairDD in comparison to other baselines in Table 22. Compared to

Table 2: Accuracy comparison on diverse IPCs.

| Methods Datasets | IPC | Random Acc. | DM Acc. | +FairDD Acc. | DC Acc. | +FairDD Acc. | IDC Acc. | +FairDD Acc. | DREAM Acc. | +FairDD Acc. | Whole Acc. |
|---|---|---|---|---|---|---|---|---|---|---|---|
| C-MNIST (FG) | 10 | 30.75 | 25.01 | **94.61** | 71.41 | **90.62** | 53.06 | **95.67** | 75.04 | **94.04** | |
| | 50 | 47.38 | 56.84 | **96.58** | 90.54 | **92.68** | 88.55 | **96.77** | 91.02 | **94.59** | 97.71 |
| | 100 | 67.41 | 78.04 | **96.79** | 91.64 | **93.23** | 90.39 | **97.11** | 88.87 | **95.16** | |
| C-MNIST (BG) | 10 | 27.95 | 23.40 | **94.88** | 65.91 | **90.84** | 62.09 | **94.84** | 79.81 | **93.54** | |
| | 50 | 45.52 | 47.74 | **96.86** | 88.53 | **92.20** | 86.14 | **95.29** | 89.24 | **93.20** | 97.80 |
| | 100 | 67.28 | 79.87 | **97.33** | 90.20 | **92.73** | 89.66 | **95.84** | 90.70 | **94.06** | |
| C-FMNIST (FG) | 10 | 32.80 | 33.35 | **77.09** | 60.77 | **76.01** | 44.08 | **79.66** | 49.72 | **77.24** | |
| | 50 | 42.48 | 49.94 | **82.11** | 69.08 | **75.83** | 64.45 | **80.80** | 65.69 | **78.79** | 82.94 |
| | 100 | 55.31 | 57.99 | **83.25** | 68.84 | **74.91** | 66.37 | **80.28** | 68.25 | **78.51** | |
| C-FMNIST (BG) | 10 | 24.96 | 22.26 | **71.10** | 47.32 | **68.51** | 37.59 | **72.67** | 45.30 | **71.56** | |
| | 50 | 34.92 | 36.27 | **79.07** | 60.58 | **75.80** | 46.20 | **73.72** | 53.62 | **72.80** | 77.97 |
| | 100 | 44.87 | 49.30 | **80.63** | 62.70 | **71.76** | 48.61 | **73.18** | 53.32 | **73.00** | |
| CIFAR10-S | 10 | 23.60 | 37.88 | **45.17** | 37.88 | **41.82** | 48.30 | **56.40** | 55.09 | **58.40** | |
| | 50 | 36.46 | 45.02 | **58.84** | 41.28 | **49.26** | 47.26 | **57.84** | 57.59 | **61.85** | 69.78 |
| | 100 | 39.34 | 48.11 | **61.33** | 42.73 | **51.74** | 47.27 | **56.98** | 57.14 | **62.70** | |
| CelebA | 10 | 54.51 | 61.79 | **64.37** | 57.19 | **57.63** | 61.49 | **63.54** | 64.38 | **66.26** | |
| | 50 | 55.99 | 64.61 | **68.50** | 60.16 | 59.89 | 60.75 | **66.89** | 64.62 | **68.26** | 74.09 |
| | 100 | 60.62 | 65.13 | **68.84** | 62.53 | 61.89 | 64.04 | **67.24** | 62.58 | **64.12** | |

Table 3: Cross-arch. comparison.

| Method | Cross arch. | DM DEO$_M$ | DM DEO$_A$ | DM Acc. | DM+FairDD DEO$_M$ | DM+FairDD DEO$_A$ | DM+FairDD Acc. |
|---|---|---|---|---|---|---|---|
| C-MNIST (FG) | ConvNet | 100.0 | 91.68 | 56.84 | **10.05** | **5.46** | 96.58 |
| | AlexNet | 100.0 | 98.82 | 44.02 | **10.35** | **6.16** | 96.12 |
| | VGG11 | 99.70 | 70.73 | 75.22 | **9.55** | **5.39** | 96.80 |
| | ResNet18 | 100.0 | 96.00 | 52.05 | **8.40** | **4.63** | 97.13 |
| | Mean | 99.93 | 89.31 | 57.03 | **9.59** | **5.41** | 96.66 |
| C-FMNIST (BG) | ConvNet | 100.0 | 99.71 | 36.27 | **24.50** | **14.47** | 79.07 |
| | AlexNet | 100.0 | 99.75 | 22.72 | **20.60** | **14.11** | 76.14 |
| | VGG11 | 100.0 | 97.77 | 43.11 | **21.60** | **14.36** | 78.57 |
| | ResNet18 | 100.0 | 99.78 | 23.37 | **22.50** | **14.96** | 75.21 |
| | Mean | 100.0 | 99.25 | 31.37 | **22.30** | **14.73** | 77.25 |
| CIFAR10-S | ConvNet | 75.13 | 55.70 | 45.02 | **18.28** | **7.35** | 58.84 |
| | AlexNet | 75.30 | 52.57 | 36.09 | **15.84** | **5.12** | 49.16 |
| | VGG11 | 61.48 | 44.05 | 43.23 | **11.51** | **4.16** | 52.65 |
| | ResNet18 | 76.23 | 54.35 | 38.03 | **16.44** | **5.14** | 50.93 |
| | Mean | 72.04 | 51.67 | 40.59 | **15.27** | **5.44** | 52.90 |
| CelebA | ConvNet | 40.26 | 38.81 | 64.61 | **14.08** | **9.87** | 68.50 |
| | AlexNet | 32.51 | 31.62 | 63.10 | **9.38** | **5.75** | 64.24 |
| | VGG11 | 26.03 | 24.63 | 61.57 | **8.95** | **6.32** | 62.05 |
| | ResNet18 | 25.60 | 24.93 | 60.32 | **6.72** | **4.29** | 61.80 |
| | Mean | 31.10 | 30.25 | 62.40 | **9.78** | **6.58** | 64.15 |

*Random*, training the model by vanilla DDs yields better performance. This shows that vanilla DDs capture the informative patterns of majority groups, improving their TA accuracy. However, by focusing on dominant patterns in majority groups, they neglect the important patterns in minority groups within the training datasets. Thus, their representation coverage is limited. In contrast, FairDD proposes synchronized matching to push the $\mathcal{S}$ to cover each group, and as a result, the generated $\mathcal{S}$ retains key patterns of all groups and achieves comprehensive coverage. For example, DM obtains 25.01 at IPC = 10 on C-MNIST (FG), and its accuracy boosts to 94.61 when applying FairDD. In real-world CelebA, FairDD obtains comparable performance for DC and presents superiority over vanilla DDs. These demonstrate that FairDD could mitigate the bias without compromising accuracy.

**Generalization to diverse architectures**   Here, we investigate the cross-model generalization of FairDD, where ConvNet is used to condense datasets, and we evaluate $\mathcal{S}$ on other architectures, including AlexNet, VGG11, and ResNet18. We compare DM and FairDD across four datasets at IPC = 50, evaluating performance against BG, FG, BG & FG, and real-world biases. As shown in Table 23, among these architectures, FairDD achieves DEO$_M$ of 10.05, 10.35, 9.55, and 8.40 on C-MNIST (FG), DEO$_A$ of 14.47, 14.11, 14.36, and 14.96 on C-FMNIST (BG), and accuracy of 58.84, 49.16, 52.65, and 50.93 on CIFAR10-S. These steady results suggest that $\mathcal{S}$ generated by FairDD are not restricted to the model used for distillation but generalize well across diverse architectures. Additionally, with the model capacity increasing, the model generally tends to be more fair to all groups. However, the accuracy sometimes decreases, such as when it drops from 58.84 (ConvNet) to 50.93 (ResNet18) in CIFAR10-S and from 68.50 (ConvNet) to 61.80 (ResNet18) in CelebA. We assume that while increased attention from larger models can lead to accuracy gains for minority groups, it may limit the representations for majority groups at certain levels. The accuracy gains for minority groups may be smaller than the accuracy losses for majority groups, particularly in larger models that have limited potential improvement in recognizing minority groups. As a result, overall accuracy may decrease despite fairness improvement.

### 5.3 Result Analysis

**Visualization analysis on representation coverage**   We investigate whether the FairDD effectively covers the whole distribution of original datasets. For this purpose, we first feed the original training set into the randomly initialized network used in the distillation, to extract the corresponding features. Subsequently, we use the same network to extract features of distilled dataset $\mathcal{S}$ from DM and FairDD. As shown in Fig. 4(a), the synthetic samples in vanilla DDs almost locate the majority group for optimizing the original alignment objective. In this case, vanilla DDs neglect to condense the key

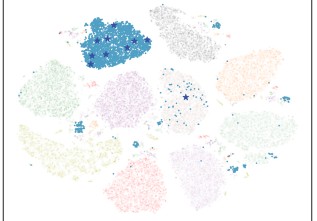 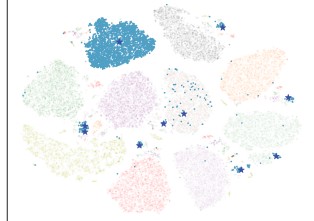

(a) The $\mathcal{S}$ distribution of generated by DM.   (b) The $\mathcal{S}$ distribution of generated by FairDD.

Figure 4: Feature coverage comparison on TA between DM and DM+FairDD. The training and synthetic dataset features are extracted by $\phi_\theta$. We visualize one class and make the remaining classes transparent for presentation. The $\mathcal{S}$ generated by DM and FairDD are marked by stars in (a) and (b).

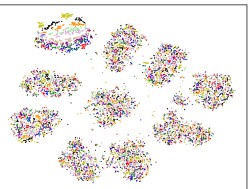 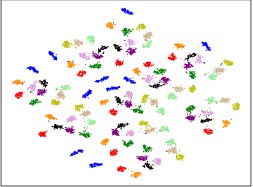 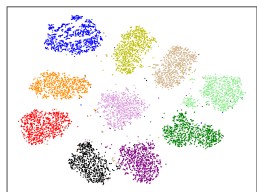

(a) PA t-SNE of DM.    (b) PA t-SNE of FairDD.    (c) TA t-SNE of DM.    (d) TA t-SNE of FairDD.

Figure 5: T-SNE visualization comparison on PA and TA between DM and DM+FairDD. Each color represents distinct PA groups in (a) and (b). (c) and (d) use different colors to represent different PA. In (a), DM shows obvious distinctiveness towards different PA, whereas in (b), DM+FairDD eliminates the recognition of PA. As illustrated in (c) and (d), DM+FairDD optimizes the model to produce compact TA representations, but DM fails to cluster TA from the same class.

Table 4: Ablation on BR at IPC = 50.

| Methods Dataset | BR | DM $DEO_M$ | DM $DEO_A$ | DM Acc. | DM+FairDD $DEO_M$ | DM+FairDD $DEO_A$ | DM+FairDD Acc. |
|---|---|---|---|---|---|---|---|
| C-MNIST (FG) | 0.85 | 99.54 | 70.13 | 76.24 | **10.13** | **5.20** | **96.62** |
| | 0.90 | 100.0 | 91.68 | 56.84 | **10.05** | **5.46** | **96.58** |
| | 0.95 | 100.0 | 100.0 | 33.73 | **10.30** | **5.84** | **96.05** |
| C-FMNIST (BG) | 0.85 | 100.0 | 95.54 | 46.14 | **23.75** | **13.85** | **79.61** |
| | 0.90 | 100.0 | 99.71 | 36.27 | **24.50** | **14.47** | **79.07** |
| | 0.95 | 100.0 | 99.79 | 26.30 | **29.15** | **17.72** | **78.46** |
| CIFAR10-S | 0.85 | 71.75 | 50.11 | 46.99 | **16.44** | **6.58** | **59.12** |
| | 0.90 | 75.13 | 55.70 | 45.02 | **18.28** | **7.35** | **58.84** |
| | 0.95 | 75.43 | 58.58 | 43.56 | **17.49** | **7.10** | **58.18** |

Table 5: Ablation on initialization at IPC = 50.

| Methods Dataset | Init. | DM $DEO_M$ | DM $DEO_A$ | DM Acc. | DM+FairDD $DEO_M$ | DM+FairDD $DEO_A$ | DM+FairDD Acc. |
|---|---|---|---|---|---|---|---|
| C-MNIST (FG) | Random | 100.0 | 91.68 | 56.84 | **10.05** | **5.46** | **96.58** |
| | Noise | 100.0 | 99.64 | 41.45 | **9.33** | **5.28** | **96.06** |
| | Hybrid | 100.0 | 99.06 | 39.97 | **9.03** | **5.33** | **96.27** |
| C-FMNIST (BG) | Random | 100.0 | 99.71 | 36.27 | **24.50** | **14.47** | **79.07** |
| | Noise | 100.0 | 99.67 | 22.92 | **23.00** | **14.40** | **78.84** |
| | Hybrid | 100.0 | 99.68 | 26.38 | **21.45** | **14.34** | **79.19** |
| CIFAR10-S | Random | 75.13 | 55.70 | 45.02 | **18.28** | **7.35** | **58.84** |
| | Noise | 55.28 | 37.26 | 46.97 | **16.15** | **6.13** | **56.41** |
| | Hybrid | 65.59 | 46.18 | 45.16 | **17.30** | **6.71** | **56.78** |

patterns of minority groups. This leads to the information loss of minority groups in $\mathcal{S}$. FairDD achieves overall coverage for both majority and minority groups in Fig. 4(b). This is because FairDD introduces synchronized matching to reformulate the distillation objective for aligning the PA-wise groups rather than being dominated by the majority group like vanilla DDs. In doing so, FairDD avoids $\mathcal{S}$ collapsing into the majority group and retains informative patterns from all groups.

**Visualization analysis on fairness and accuracy** To intuitively present the effectiveness of FairDD, we train $g_\psi$ using $\mathcal{S}$ of C-MNIST (FG) distilled by DM and FairDD, and then extract the features from the test dataset. Different colors paint these resulting features according to PA and TA, respectively. As shown in Figs. 5(a) and 5(b), features with the same PA tend to form a cluster, indicating that the model trained on DM is sensitive to PA and thus failing to guarantee fairness among all PA. In contrast with DM, the feature distributions in Fig. 5(b) exhibit nearly complete overlaps across all PA. It shows that the model trained on FairDD is agnostic to PA and does not exhibit bias towards these PA. Besides the PA fairness, we also study the feature distribution from the TA perspective. Fig. 5(c) shows that features belonging to one TA scatter and fail to provide compact representations for one class. The failure of DM can be attributed to model bias toward PA. Combined with Fig. 5(a), it can be observed that PA has a stronger influence on the feature distribution compared to TA. As a result, PA-wise representations are tightly clustered, but representations from the same TA are divided into PA-wise parts. In contrast, FairDD proposes synchronized matching effectively mitigates this by treating each PA group equally within one TA. The equal treatment allows different PA groups within the same TA to cluster more easily, leading to more compact representations that benefit capturing class semantics in Fig. 5(d). These results highlight the superiority of FairDD in improving PA fairness and TA accuracy. We also provide visualization analysis on $\mathcal{S}$ generation in Appendix M. Additional computation overhead is provided in Appendix C.

## 5.4 ABLATION STUDY

**Ablation on biased ratio of original datasets** BR reflects the extent of unfairness in the original datasets and indicates the level of PA skew that the distillation process of $\mathcal{S}$ will encounter. We investigate the impact of BR values on fairness performance by setting BR to $\{0.85, 0.90, 0.95\}$ on C-MNIST (FG), C-FMNIST (BG), and CIFAR10-S. The results at IPC = 50 in Table 24 show that DM is sensitive to the BR of original datasets, with its $DEO_M$ decreasing from 70.13 to 100.0 as BR increases from 0.85 to 0.95. A similar trend is observed in other datasets. Compared to DM, FairDD maintains consistent fairness and accuracy levels across different biases. This is attributed to the synchronized matching, which explicitly aligns each PA-wise subtarget, reducing sensitivity to group-specific sample numbers. This shows FairDD's robustness to PA skew in the original datasets.

**Ablation on initialization of synthetic images** The initialization of $\mathcal{S}$ determines the prior information obtained by DDs. We examine the effect of different initialization using three strategies: *random*: randomly drawing samples from the original datasets to initialize $\mathcal{S}$; *Noise*: using noise obeying the standard normal distribution for initialization; and *hybrid*: selecting each initialization sample with equal probability to follow either the *random* or *Noise* strategy. In Table 25, the $\text{DEO}_M$ and $\text{DEO}_A$ metrics of DM have largely fluctuate under these initialization strategies. This suggests that DM fails to incorporate sufficient distilled information into $\mathcal{S}$, making it overly dependent on the prior information of $\mathcal{S}$. In contrast, FairDD achieves robust performance in both fairness and accuracy, indicating that the informative patterns during distillation dominate the generation of $\mathcal{S}$.

## 6 RELATED WORK

**Dataset distillation** Dataset distillation has been broadly applied to many important fields (Lee et al., 2023; He et al., 2024; Feng et al., 2023; Chen et al., 2023). The first work (Wang et al., 2018) attempts to formulate dataset distillation as a bi-level optimization problem. However, the two folds of the optimization process are time-consuming. Neural tangent kernel (Jacot et al., 2018) are utilized to obtain the close form of the inner loop (Nguyen et al., 2021; Loo et al., 2022; Zhou et al., 2022). Some works propose surrogate objectives to achieve comparable even better performance, including matching-based methods like GM (Zhao et al., 2020; Zhao & Bilen, 2021; Lee et al., 2022a), DM (Zhao & Bilen, 2023; Wang et al., 2022; Zhao & Bilen, 2022), TM (Cazenavette et al., 2022; Cui et al., 2022), soft label learning (Bohdal et al., 2020; Sucholutsky & Schonlau, 2021) and factorization (Kim et al., 2022; Deng & Russakovsky, 2022; Liu et al., 2022; Lee et al., 2022a). Cui et al. (2024) and Zhao et al. (2024) focus on reducing the bias to improve the classification performance (Vahidian et al., 2024; Wang et al., 2024). In summary, current DD approaches only pursue classification accuracy that models trained on synthetic datasets, while they neglect fairness concerning PA. Therefore, we propose FairDD to improve fairness without sacrificing TA accuracy.

**Visual fairness** With the advancement of computer vision, fair predictions without discrimination towards minority groups have become a crucial topic (Caton & Haas, 2024). According to the stage of bias mitigation, the research field of fairness algorithm (Bellamy et al., 2019) can be classified into three branches: Pre-processing (Creager et al., 2019; Louizos et al., 2015; Quadrianto et al., 2019; Sattigeri et al., 2019), In-processing (Agarwal et al., 2018; Jiang & Nachum, 2020; Zafar et al., 2017; Zhang et al., 2018; Jung et al., 2021; Wang et al., 2020; Jung et al., 2022; Zemel et al., 2013), and Post-processing (Alghamdi et al., 2020; Hardt et al., 2016). Our research falls within Pre-processing branch. Pre-processing aims to generate a fair version of datasets for downstream tasks. Several related works frame this issue as a data-to-data translation problem, utilizing generative models to produce fairer datasets concerning protected groups (Sattigeri et al., 2019; Quadrianto et al., 2019). However, unlike the traditional fairness approach that primarily focuses on reducing bias in original datasets Subramanian et al. (2021); Tarzanagh et al. (2023); Vogel et al. (2020); Rangwani et al. (2022), our work aims to integrate fairness into DD. Our objective is to mitigate the bias of original datasets while simultaneously condensing them into smaller and more informative counterparts.

## 7 CONCLUSION

This paper reveals for the first time that vanilla DDs fail to mitigate the bias of original datasets and even exacerbate the bias. To address the problem, we propose a unified fair dataset distillation framework called FairDD, broadly applicable to various DDs in DMF. FairDD requires no modifications to the architectures of vanilla DDs and introduces an easy-to-implement yet effective attribute-wise matching. This method mitigates the dominance of the majority group and ensures that synthetic datasets equally incorporate representative patterns with all protected attributes from both majority groups and minority groups. By doing so, FairDD guarantees the fairness of synthetic datasets while maintaining their representativeness for image recognition. We provide extensive theoretical analysis and empirical results to demonstrate the superiority of FairDD.

**Limitation** Since FairDD relies on PA's prior information to conduct attribute-wise matching, it is valuable to explore the scenario where PA is unavailable Liu et al. (2021). A potential solution is to generate pseudo-labels to guide FairDD through self-supervised learning or unsupervised learning.

**Board impact** This paper aims to improve data efficiency and enhance model fairness in modern machine learning, fully compliant with legal regulations. Since training a fair model from scripts with extensive data is time-consuming, our work in providing a fair condensed dataset for effective model training can have significant societal impacts. We hope our research raises attention to achieve both fairness and accuracy for dataset distillation in academia and industry.

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

Table 6: Statistics for all datasets used in our paper.

| Datasets | TA | PA | TA number | PA number | Training set size | Test set size | BR in Training set | BR in Test set | Condensed ratio | | |
|---|---|---|---|---|---|---|---|---|---|---|---|
| | | | | | | | | | 10 | 50 | 100 |
| C-MNIST (FG) | Digital number | Digital color | 10 | 10 | 60000 | 10000 | 0.90 | balance | 0.17% | 0.83% | 1.67% |
| C-MNIST (BG) | Digital number | Background color | 10 | 10 | 60000 | 10000 | 0.90 | balance | 0.17% | 0.83% | 1.67% |
| C-FMNIST (FG) | Object category | Object color | 10 | 10 | 60000 | 10000 | 0.90 | balance | 0.17% | 0.83% | 1.67% |
| C-FMNIST (BG) | Object category | Background color | 10 | 10 | 60000 | 10000 | 0.90 | balance | 0.17% | 0.83% | 1.67% |
| CIFAR10-S | Object category | Grayscale or not | 10 | 2 | 50000 | 20000 | 0.90 | balance | 0.20% | 1.00% | 2.00% |
| CelebA | Attractive | Gender | 2 | 2 | 162770 | 7656 | class0: 0.62 class1: 0.77 | balance | 0.012% | 0.061% | 0.12% |

# A  DATASEST STATISTICS

In this section, we provide detailed statistics for all datasets used in the manuscript for reproduction. As shown in Table 6, we present the target attribute (TA), protected attribute (PA), the sample number of the training set, the sample number of the test set, and the BR in the training set. Additionally, all test sets are balanced, with equal sample sizes across groups. We also report the condensed ratio at IPC 10, 50, and 100, which is computed by the ratio of the condensed dataset size to the training set size.

# B  MORE ATTRIBUTES ANALYSIS ON CELEBA

We explore additional facial attributes in CelebA to further demonstrate the robustness of FairDD. To this end, we regard `gender` as the PA, and `young`, `big_nose`, and `blond_hair` as the TA, which results in CelebA$_y$, CelebA$_b$, CelebA$_h$ and respectively. We also replace the target attribute from `attribute` to `blond_hair`, resulting in CelebA$^h$. The performance is reported in Table 7. FairDD exhibits fairer behavior compared to vanilla DDs across these attributes while maintaining comparable accuracy, as seen in Table 8.

Table 7: Fairness comparison on different attributes.

| Methods Dataset | IPC | DM DEO$_M$ | DEO$_A$ | DM+FairDD DEO$_M$ | DEO$_A$ | DC DEO$_M$ | DEO$_A$ | DC+FairDD DEO$_M$ | DEO$_A$ | Whole DEO$_M$ | DEO$_A$ |
|---|---|---|---|---|---|---|---|---|---|---|---|
| CelebA$_y$ | 10 | 34.18 | 31.49 | **13.30** | **10.38** | 20.58 | 19.26 | **10.86** | **8.55** | 25.40 | 16.02 |
| | 50 | 46.90 | 41.13 | **12.90** | **8.21** | 27.98 | 25.18 | **14.69** | **11.26** | | |
| | 100 | 44.96 | 37.84 | **9.17** | **5.11** | 27.76 | 24.26 | **19.03** | **13.61** | | |
| CelebA$_b$ | 10 | 45.57 | 45.13 | **15.63** | **13.47** | 18.17 | 16.81 | **7.54** | **6.34** | 34.48 | 25.50 |
| | 50 | 51.91 | 51.13 | **14.44** | **12.01** | 23.85 | 22.34 | **20.58** | **16.87** | | |
| | 100 | 52.75 | 51.27 | **8.03** | **6.10** | 24.48 | 23.53 | **12.15** | **11.00** | | |
| CelebA$_h$ | 10 | 17.01 | 9.56 | **7.76** | **6.02** | 12.44 | 8.01 | **9.25** | **7.31** | 15.53 | 11.56 |
| CelebA$^h$ | 10 | 30.28 | 20.76 | **12.70** | **8.28** | 25.94 | 15.11 | **16.78** | **9.88** | 46.67 | 26.11 |

Table 8: Accuracy comparison.

| Methods Dataset | IPC | DM Acc. | +FairDD Acc. | DC Acc. | +FairDD Acc. | Whole Acc. |
|---|---|---|---|---|---|---|
| CelebA$_y$ | 10 | 62.34 | **63.79** | 55.91 | **56.99** | 75.99 |
| | 50 | 63.59 | **67.33** | **59.87** | 59.42 | |
| | 100 | 66.68 | **69.90** | **63.53** | 61.59 | |
| CelebA$_b$ | 10 | 57.46 | **59.50** | 52.91 | **54.67** | 66.80 |
| | 50 | 58.71 | **62.39** | **56.55** | 55.46 | |
| | 100 | 60.30 | **64.34** | **57.65** | 57.15 | |
| CelebA$_h$ | 10 | 63.64 | **64.86** | **58.04** | 57.55 | 75.33 |
| CelebA$^h$ | 10 | 77.66 | **79.71** | 72.07 | **75.03** | 79.44 |

To further study the generalization of FairDD, we regard `blond_hair` as the protected attribute and `attractive` as the target attribute, resulting in CelebA$_b$. As illustrated in Table, FairDD+DM obtains 7.76% DEO$_M$ and 6.02% DEO$_A$, outperforming DM by 9.25% and 3.54%. Accuracy has also been improved.

# C  COMPUTATION OVERHEAD

In this section, we investigate the computational efficiency of FairDD. Since FairDD performs fine-grained alignment at the group level, we evaluate the impact of the number of groups on training time (min) and peak GPU memory consumption (MB). As shown in Table 9, FairDD requires more time than vanilla DDs on C-MNIST (FG), and the time increases as the number of groups (PA) grows. This phenomenon is particularly noticeable in DC because the gradient must be computed with respect to the number of groups. In contrast, DM avoids computing gradients independently for each group and directly computes all embeddings once in a single pass, reducing the additional overhead caused by FairDD's group-level alignment. Regarding GPU memory usage, FairDD incurs no obvious additional overhead compared to vanilla DDs.

Here, we further supplement the overhead analysis with respect to image resolutions. We conduct experiments on CMNIST, CelebA (32), CelebA (64), and CelebA (96) on DM and DC at IPC=10.

Table 9: Comparison of computation overhead on FairDD and vanilla DDs.

| Group | 0 (vanilla DD) | | 2 (FairDD) | | 4 (FairDD) | | 6 (FairDD) | | 8 (FairDD) | | 10 (FairDD) | |
|---|---|---|---|---|---|---|---|---|---|---|---|---|
| number | T (min) | G (MB) | T (min) | G (MB) | T (min) | G (MB) | T (min) | G (MB) | T (min) | G (MB) | T (min) | G (MB) |
| DC | 70 | 2143 | 94 | 2345 | 128 | 2369 | 152 | 2393 | 181.8 | 2419 | 210 | 2443 |
| DM | 26.2 | 1579 | 31.75 | 1579 | 33.2 | 1579 | 35.2 | 1579 | 36.5 | 1579 | 36.9 | 1579 |

DM and DC align different signals, which would bring different effects. As illustrated in Table 10, it can be observed that FairDD + DM does not require additional GPU memory consumption but does necessitate more time. The time gap increases from 0.42 minutes to 1.79 minutes as input resolution varies (e.g., CelebA 32 × 32, CelebA 64 × 64, and CelebA 96 × 96); however, the gap remains small. This can be attributed to FairDD performing group-level alignment on features, which is less influenced by input resolution. Notably, although CMNIST and CelebA (32 × 32) share the same resolution, the time gap is more pronounced for CMNIST (e.g., 3 minutes). This is attributed to CMNIST having 10 attributes, whereas CelebA (32 × 32) has only 2 attributes. These indicate that FairDD + DM requires no additional GPU memory consumption. Its additional time depends on both input resolution and the number of groups, but the number of groups more significantly influences it. As for DC, FairDD requires additional GPU memory and time. Since FairDD + DC explicitly computes group-level gradients, the resulting gradient caches cause FairDD + DC to consume more memory. The additional consumption is acceptable compared to the performance gain on fairness. Additionally, the time gap is relatively larger than that observed between DM and FairDD + DM. Similar to DM, the group number is the primary factor contributing to additional time consumption compared to input resolution.

Table 10: Comparison of computation overhead for IPC = 10.

| Methods | Group | DM | | DM+FairDD | | DC | | DC+FairDD | |
|---|---|---|---|---|---|---|---|---|---|
| Dataset | number | Time | Memory | Time | Memory | Time | Memory | Time | Memory |
| CMNIST | 10 | 15.55 | 1227 | 18.55 | 1227 | 58.75 | 1767 | 83.13 | 1893 |
| CelebA32 × 32 | 2 | 10.93 | 2293 | 11.35 | 2293 | 32.98 | 2413 | 34.65 | 2479 |
| CelebA64 × 64 | 2 | 11.18 | 8179 | 12.20 | 8177 | 43.67 | 8525 | 47.07 | 8841 |
| CelebA96 × 96 | 2 | 12.83 | 17975 | 14.62 | 17975 | 82.37 | 18855 | 86.88 | 19437 |

# D  ADDITIONAL EXPERIMENT ON UTKFACE

We also conduct another real image dataset namely UTKFace, commonly used for fairness. It consists of 20,000 face images including three attributes, age, gender, and race. We follow a common setting and treat age as the target attribute and gender as the protected attribute. We test DM and FairDD + DM with the same parameters, the results in Table 11 show that our method outperforms the vanilla dataset distillation by 16.1% and 8.92% on the $DEO_M$ and $DEO_A$. Similar results are observed at IPC = 50.

Table 11: UTKFace performance on DM.

| Methods | IPC | DM | | | DM+FairDD | | |
|---|---|---|---|---|---|---|---|
| Dataset | | Acc. | $DEO_M$ | $DEO_A$ | Acc. | $DEO_M$ | $DEO_A$ |
| UTKFace | 10 | 66.67 | 32.97 | 16.31 | 67.72 | 16.87 | 7.39 |
| | 50 | 73.15 | 28.58 | 14.03 | 74.66 | 10.59 | 5.09 |

# E  ABLATION STUDY ON WEIGHTING MECHANISM

We denote the model with inversely proportional weighting as $FairDD_{inverse}$. Our experiments on C-FMNIST and CIFAR10-S at IPC=10 reveal that $FairDD_{inverse}$ suffers significant performance degradation, with $DEO_M$ increasing from 33.05% to 56.60% and $DEO_A$ rising from 19.72% to 35.13% in terms of fairness performance metrics. Additionally, there is a decline in accuracy for TA.

| Methods Dataset | IPC | DM | | | DM+FairDD$_{inverse}$ | | | DM+FairDD | | |
|---|---|---|---|---|---|---|---|---|---|---|
| | | Acc. | DEO$_M$ | DEO$_A$ | Acc. | DEO$_M$ | DEO$_A$ | Acc. | DEO$_M$ | DEO$_A$ |
| C-FMNIST (BG) | 10 | 22.26 | 100.0 | 99.05 | 70.55 | 56.60 | 35.13 | 71.10 | 33.05 | 19.72 |
| CIFAR10-S | 10 | 37.88 | 59.20 | 39.31 | 38.14 | 48.27 | 37.41 | 45.17 | 31.75 | 8.73 |

Table 12: Performance between FairDD and FairDD$_{inverse}$.

We attribute this degradation to the excessive penalization of groups with larger sample sizes. The success of FairDD lies in grouping all samples with the same PA into a single group and performing the group-level alignment. Each group contributes equally to the total alignment, inherently mitigating the effects of imbalanced sample sizes across different groups. However, penalizing groups based on sample cardinality reintroduces an unexpected bias related to group size in the information condensation process. This results in large groups receiving smaller weights during alignment, placing them in a weaker position and causing synthetic samples to deviate excessively from large (majority) groups. Consequently, majority patterns become underrepresented, ultimately hindering overall performance.

## F  ABLATION STUDY ON GROUP LABEL NOISE

Here, we evaluate the robustness of spurious group labels could provide more insights. We randomly sample the entire dataset according to a predefined ratio. These samples are randomly assigned to group labels to simulate noise. To ensure a thorough evaluation, we set sample ratios at 10%, 15%, 20%, and 50%. As shown in the table, when the ratio increases from 10% to 20%, the DEO$_M$ results range from 14.93% to 18.31% with no significant performance variations observed. These results indicate that FairDD is robust to noisy group labels. However, as the ratio increases further to 50%, relatively significant performance variations become apparent. It can be understood that under a high noise ratio, the excessive true samples of majority attributes are assigned to minority labels. This causes the minority group center to shift far from its true center and thus be underrepresented.

Table 13: Ablation study on group label noise.

| Methods Dataset | IPC | DM | | | DM+FairDD | | | DM+FairDD (10%) | | | DM+FairDD (15%) | | | DM+FairDD (20%) | | | DM+FairDD (50%) | | |
|---|---|---|---|---|---|---|---|---|---|---|---|---|---|---|---|---|---|---|---|
| | | Acc. | DEO$_M$ | DEO$_A$ | Acc. | DEO$_M$ | DEO$_A$ | Acc. | DEO$_M$ | DEO$_A$ | Acc. | DEO$_M$ | DEO$_A$ | Acc. | DEO$_M$ | DEO$_A$ | Acc. | DEO$_M$ | DEO$_A$ |
| CMNIST (BG) | 10 | 27.95 | 100.0 | 99.11 | 94.88 | 13.42 | 6.77 | 94.34 | 16.54 | 7.81 | 94.44 | 17.90 | 8.61 | 94.32 | 18.31 | 9.20 | 89.56 | 66.19 | 25.97 |

## G  ABLATION STUDY ON BALANCED ORIGINAL DATASET.

We synthesized a fair version of CelebA, referred to as CelebA$_{Fair}$. The target attribute is attractive (attractive and unattractive), and the protected attribute is gender (female and male). In the original dataset, the sample numbers for female-attractive, female-unattractive, male-attractive, and male-unattractive groups are imbalanced. To create a fair version, CelebA$_{Fair}$ samples the number of instances based on the smallest group, ensuring equal representation across all four groups. We tested the fairness performance of FairDD and DM at IPC = 10, as well as the performance of models trained on the full dataset. As shown in Table 14, vanilla DD achieves 14.33% DEO$_A$ and 8.77% DEO$_M$. In comparison, the full dataset achieves 3.66% DEO$_A$ and 2.77% DEO$_M$. While DM still exacerbates bias with a relatively small margin, this is primarily due to partial information loss introduced during the distillation process. FairDD produces fairer results, achieving 11.11% DEO$_A$ and 6.68% DEO$_M$.

Table 14: Performance on balanced original dataset.

| Methods Dataset | IPC | Whole | | | DM | | | DM+FairDD | | |
|---|---|---|---|---|---|---|---|---|---|---|
| | | Acc. | $DEO_M$ | $DEO_A$ | Acc. | $DEO_M$ | $DEO_A$ | Acc. | $DEO_M$ | $DEO_A$ |
| $CelebA_{Fair}$ | 10 | 76.33 | 3.66 | 2.77 | 63.31 | 14.33 | 8.77 | 63.17 | 11.11 | 6.68 |

# H   ABLATION STUDY ON NUANCED PA GROUPS

We perform a fine-grained PA division. For example, we consider gender and wearing-necktie as two correlated attributes and divide them into four groups: males with a necktie, males without a necktie, females with a necktie, and females without a necktie ($CelebA_{g\&n}$). Similarly, we consider gender and paleskin and divide them into four groups ($CelebA_{g\&p}$). Their target attribute is attractive. As shown in the Table 15, FairDD outperforms vanilla DD in the accuracy and fairness performance on these two experiments. The performance for necktie and gender is improved from 57.50% to 25.00% on $DEO_M$ and 52.79% to 21.73% on $DEO_A$. Accuracy is also improved from 63.25% to 67.98%. Similar results can be observed for gender and paleskin. Hence, FairDD can mitigate more fine-grained attribute bias, even when there is an intersection between attributes.

Table 15: Performance on nuanced groups.

| Methods Dataset | IPC | DM | | | DM+FairDD | | |
|---|---|---|---|---|---|---|---|
| | | Acc. | $DEO_M$ | $DEO_A$ | Acc. | $DEO_M$ | $DEO_A$ |
| $CelebA_{g\&n}$ | 10 | 63.25 | 57.50 | 52.79 | 67.98 | 25.00 | 21.73 |
| $CelebA_{g\&p}$ | 10 | 62.48 | 44.81 | 41.60 | 64.37 | 26.92 | 19.33 |

# I   ABLATION STUDY ON IMBALANCED PA GROUPS

To further study FairDD robustness under more biased scenarios, we keep the sample number of the majority group in each class invariant and allocate the sample size to the remaining 9 minority groups with increasing ratios, i.e., 1:2:3:4:5:6:7:8:9. We denote this variant $CMNIST_{unbalance}$ This could help create varying extents of underrepresented samples for different minority groups. Notably, the least-represented PA groups account for only about 1/500 of the entire dataset, which equates to just 12 samples out of 6000 in $CMNIST_{unbalance}$. As shown in Table 16, FairDD achieves a robust performance of 16.33% $DEO_M$ and 9.01% $DEO_A$ compared to 17.04% and 7.95% in the balanced PA groups. A similar steady behavior is observed in accuracy, which changes from 94.45% to 94.61%. This illustrates the robustness of FairDD under different levels of dataset imbalance.

Table 16: Performance on imbalanced PA.

| Methods Dataset | IPC | DM | | | DM+FairDD | | |
|---|---|---|---|---|---|---|---|
| | | Acc. | $DEO_M$ | $DEO_A$ | Acc. | $DEO_M$ | $DEO_A$ |
| CMNIST | 10 | 25.01 | 100.0 | 99.96 | 94.61 | 17.04 | 7.95 |
| $CMNIST_{unbalance}$ | 10 | 23.38 | 100.0 | 99.89 | 94.45 | 16.33 | 9.01 |

# J   EXPLORATION ON VISION TRANSFORMER AS BACKBONE

Although the Vision Transformer (ViT) is a powerful backbone network, to the best of my knowledge, current DDs, such as DM and DC, have not yet utilized ViT as the extraction network. We conducted experiments using 1-layer, 2-layer, and 3-layer ViTs. As shown in Table 17, vanilla DM at IPC=10 suffers performance degradation in classification, dropping from 25.01% to 18.63%. Moreover, as the number of layers increases, the performance deteriorates more severely. This suggests that current DDs are not directly compatible with ViTs. While FairDD still outperforms DM in both accuracy

and fairness metrics, the observed improvement gain is smaller compared to results obtained on convolutional networks. Further research into leveraging ViTs for DD and FairDD is a promising direction worth exploring.

Table 17: Exploration on ViT architecture.

| Methods Dataset | IPC | DM | | | DM+FairDD | | |
|---|---|---|---|---|---|---|---|
| | | Acc. | $DEO_M$ | $DEO_A$ | Acc. | $DEO_M$ | $DEO_A$ |
| ViT1 | 10 | 18.63 | 100.0 | 98.48 | 56.15 | 82.10 | 56.72 |
| ViT2 | 10 | 18.28 | 100.0 | 98.99 | 33.89 | 72.85 | 40.97 |
| ViT3 | 10 | 16.15 | 100.0 | 95.75 | 26.70 | 65.71 | 29.46 |

## K EXPLORATION ON CHALLENGING DATASET

We created CIFAR100-S following the same operation as CIFAR10-S, where the grayscale or not is regarded as PA. Due to the time limit, we supplemented CIFAR100-S on DM at IPC=10. As shown in Table 18, DM achieves the classification accuracy of 22.84%, and the fairness of 69.9% $DEO_M$ and 25.37% $DEO_A$. Compared to vanilla DM, FairDD obtains more accurate classification performance and mitigates the bias to the minority groups, with 27.30% $DEO_M$ and 15.54% $DEO_A$ improvement.

Table 18: Exploration on challenging dataset.

| Methods Dataset | IPC | DM | | | DM+FairDD | | |
|---|---|---|---|---|---|---|---|
| | | Acc. | $DEO_M$ | $DEO_A$ | Acc. | $DEO_M$ | $DEO_A$ |
| CIFAR100-S | 10 | 19.69 | 69.90 | 25.37 | 22.84 | 42.60 | 9.83 |

## L COMPARISON WITH MTT

Unlike DMF, MTT uses a two-stage method to condense the dataset. First, it stores the model trajectories, and then it uses these trajectories to guide the generation of the synthetic dataset. To provide a comprehensive comparison, we compare FairDD with MTT, as shown in Tables 19 and 20.

Table 19: Fairness comparison on diverse IPCs. The best results are highlighted in bold.

| Methods Dataset | IPC | Random | | MTT | | DM | | DM+FairDD | | Whole | |
|---|---|---|---|---|---|---|---|---|---|---|---|
| | | $DEO_M$ | $DEO_A$ | $DEO_M$ | $DEO_A$ | $DEO_M$ | $DEO_A$ | $DEO_M$ | $DEO_A$ | $DEO_M$ | $DEO_A$ |
| C-MNIST (FG) | 10 | 100.0 | 98.72 | 25.70 | 14.86 | 100.0 | 99.96 | **17.04** | **7.95** | | |
| | 50 | 100.0 | 99.58 | 25.46 | 12.60 | 100.0 | 91.68 | **10.05** | **5.46** | 10.10 | 5.89 |
| | 100 | 100.0 | 88.64 | 26.81 | 13.02 | 99.36 | 66.38 | **8.17** | **4.86** | | |
| C-FMNIST (BG) | 10 | 100.0 | 99.40 | 97.00 | 62.46 | 100.0 | 99.68 | **33.05** | **19.72** | | |
| | 50 | 100.0 | 98.52 | 96.60 | 62.02 | 100.0 | 99.71 | **24.50** | **14.47** | 91.40 | 51.68 |
| | 100 | 100.0 | 96.05 | 97.20 | 63.66 | 100.0 | 93.88 | **21.95** | **13.33** | | |

Table 20: Accuracy comparison on diverse IPCs.

| Methods Dataset | IPC | Random Acc. | MTT Acc. | DM Acc. | +FairDD Acc. | Whole Acc. |
|---|---|---|---|---|---|---|
| C-MNIST (FG) | 10 | 30.75 | 92.00 | 25.01 | **94.61** | |
| | 50 | 47.38 | 94.08 | 56.84 | **96.58** | 97.71 |
| | 100 | 67.41 | 94.29 | 78.04 | **96.79** | |
| C-FMNIST (BG) | 10 | 24.96 | 67.92 | 22.26 | **71.10** | |
| | 50 | 34.92 | 70.32 | 36.27 | **79.07** | 77.97 |
| | 100 | 44.87 | 70.74 | 49.30 | **80.63** | |

From the results, FairDD outperforms MTT in both fairness and accuracy. Notably, MTT surpasses DM by a large margin, which we attribute to two factors: 1) Unlike DMF, which is directly influenced by biased data, MTT aligns the model parameters to optimize the synthetic dataset, and this indirect alignment reduces the impact of bias in the data. 2) An accurate model typically conceals its inherent unfairness, as it can better classify each class despite underlying biases. For example, when *Whole* model achieves high accuracy on the C-MNIST (FG) dataset, MTT inherits this accuracy and conceals its biases. However, when the model's accuracy declines on the C-FMNIST (BG) dataset, MTT reveals its underlying unfairness in Fig. 6(a). In contrast, FairDD directly addresses unfairness rather than relying on high accuracy to obscure biased behavior in Fig. 6(b).

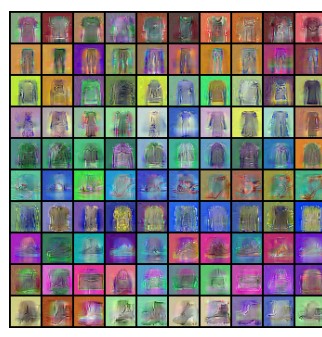

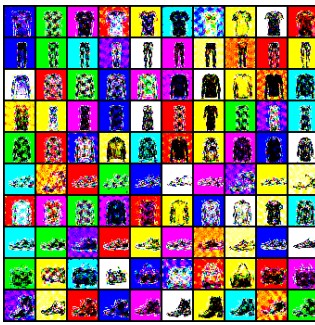

(a) MTT visualization.                    (b) FairDD visualization.

Figure 6: Visualization comparison on C-FMNIST (BG) between MTT and FairDD + DM.

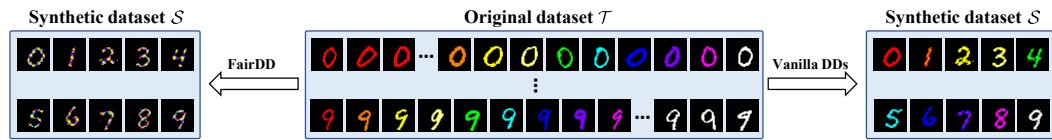

Figure 7: Visualization on $\mathcal{S}$ at IPC=1 for FairDD and vanilla DDs. **Left** is the condensed dataset using FairDD, which incorporates different PA, i.e., foreground colors. **Right** is the condensed dataset using vanilla DDs, where each class presents the same color as the corresponding majority group.

## M  ADDITIONAL VISUALIZATION ANALYSIS

**Visualization analysis on $\mathcal{S}$ generation**    We aim to investigate whether FaiDD renders the expectation of $\mathcal{S}$ locate the center among all groups, as clarified in Eq. 8. If the clarification holds, $\mathcal{S}$ should contain all PA at IPC = 1 because the expectation of $\mathcal{S}$ is equal to $\mathcal{S}$ when IPC =1. We visualize $\mathcal{S}$ at IPC=1 on C-MNIST (FG), where each class (digital number) is dominated by one color, and the rest is colored by the rest nine colors. As shown in Fig. 7, the $\mathcal{S}$ generated by FairDD combines all colors from PA groups. This suggests that FairDD can effectively incorporate all PA into resulting $\mathcal{S}$, indirectly validating the Theorem 4.1. Meanwhile, we observe that the majority groups dominate vanilla DDs according to Eq. 6, where the resulting $\mathcal{S}$ contains the colors from the corresponding majority groups.

## N  MORE VISUALIZATIONS

We provide more visualizations at IPC = 50 on different datasets in Figures 8, 9, 10, 11, and 13.

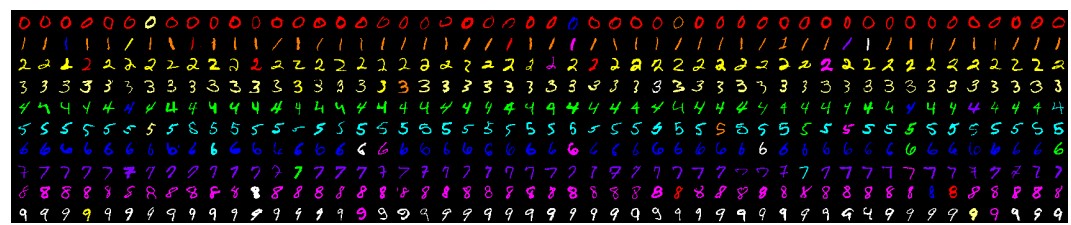

(a) Visualization of the initialized dataset at IPC = 50 in C-MNIST (FG). The foreground of each class is dominated by one color.

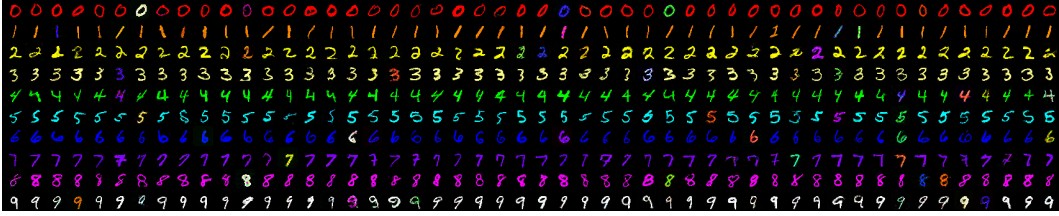

(b) Visualization of the condensed dataset at IPC = 50 in C-MNIST (FG) using Vanilla DM. The foreground of each class inherits the bias.

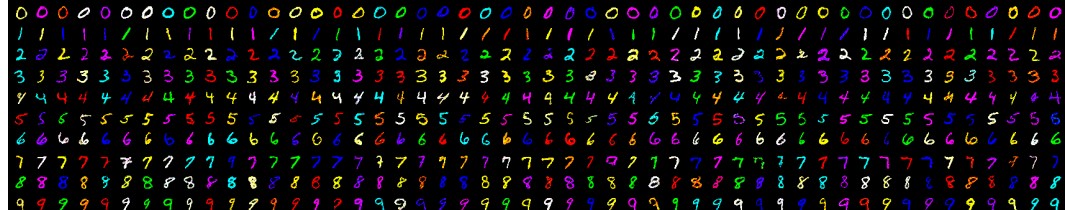

(c) Visualization of the condensed dataset at IPC = 50 in C-MNIST (FG) using FairDD + DM. The foreground of each class mitigates such bias.

Figure 8: Visualization comparison on C-MNIST (FG) between vanilla DM and FairDD + DM.

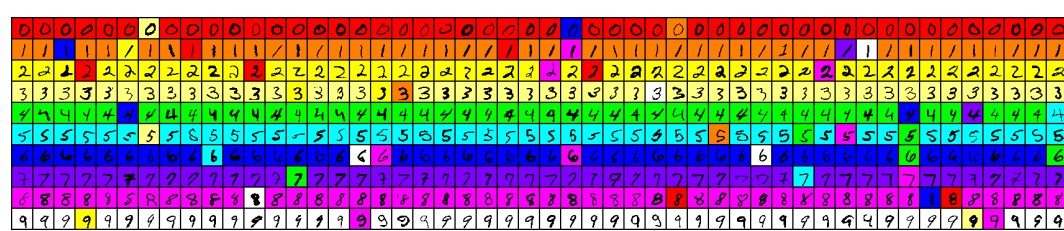

(a) Visualization of the initialized dataset at IPC = 50 in C-MNIST (BG). The background of each class is dominated by one color.

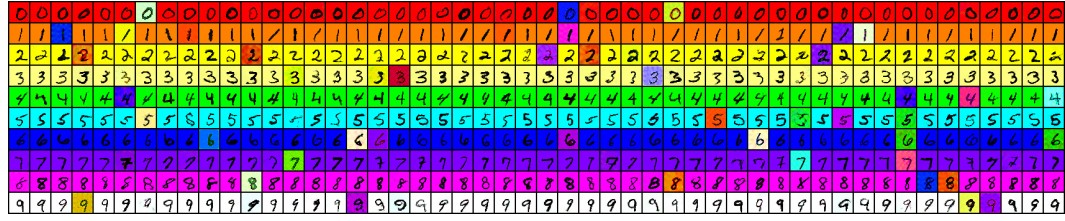

(b) Visualization of the condensed dataset at IPC = 50 in C-MNIST (BG) using Vanilla DM. The background of each class inherits the bias.

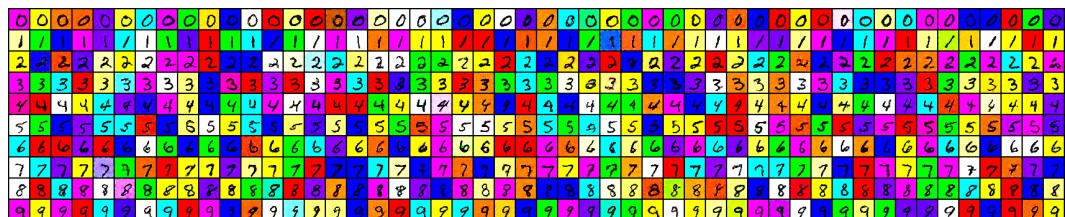

(c) Visualization of the condensed dataset at IPC = 50 in C-MNIST (BG) using FairDD + DM. The background of each class mitigates such bias.

Figure 9: Visualization comparison on C-MNIST (BG) between vanilla DM and FairDD + DM.

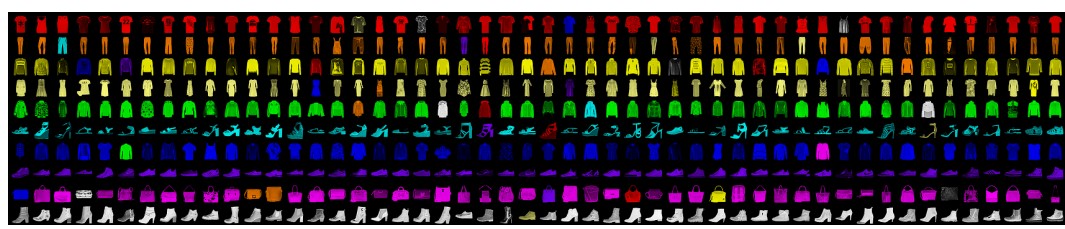

(a) Visualization of the initialized dataset at IPC = 50 in C-FMNIST (FG). The foreground of each class is dominated by one color.

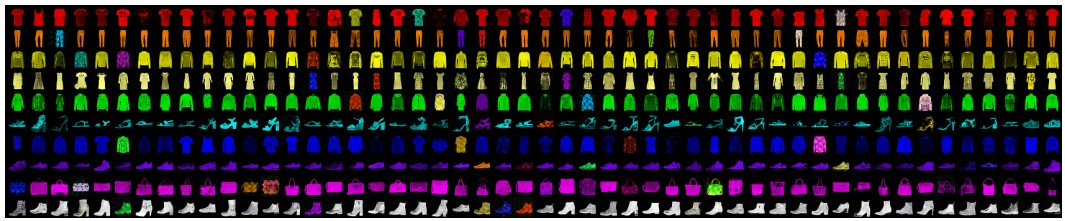

(b) Visualization of the condensed dataset at IPC = 50 in C-FMNIST (FG) using Vanilla DM. The foreground of each class inherits the bias.

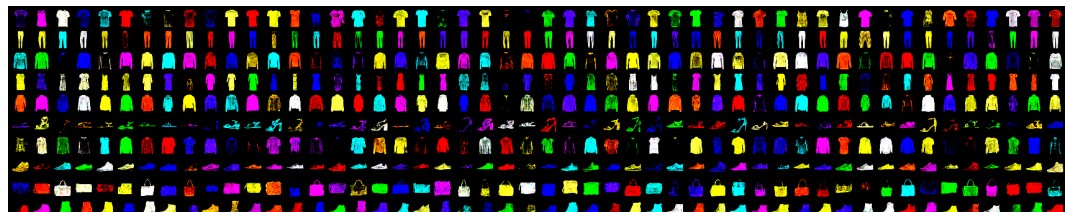

(c) Visualization of the condensed dataset at IPC = 50 in C-FMNIST (FG) using FairDD + DM. The foreground of each class mitigates such bias.

Figure 10: Visualization comparison on C-FMNIST (FG) between vanilla DM and FairDD + DM.

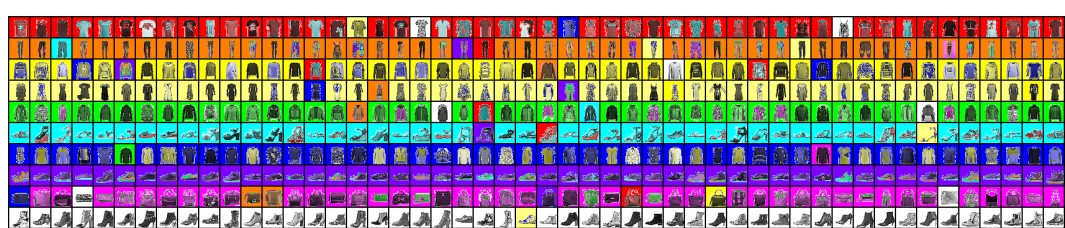

(a) Visualization of the initialized dataset at IPC = 50 in C-FMNIST (BG). The background of each class is dominated by one color.

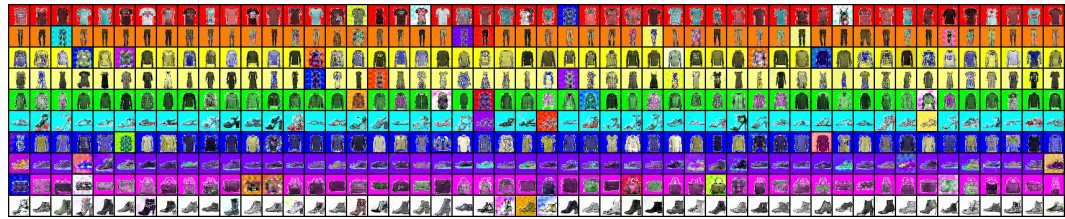

(b) Visualization of the condensed dataset at IPC = 50 in C-FMNIST (BG) using Vanilla DM. The background of each class inherits the bias.

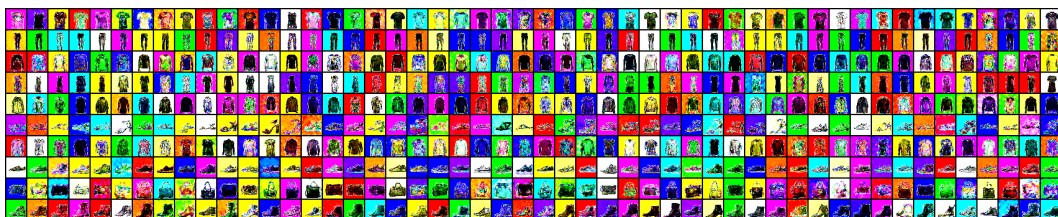

(c) Visualization of the condensed dataset at IPC = 50 in C-FMNIST (BG) using FairDD + DM. The background of each class mitigates such bias.

Figure 11: Visualization comparison on C-FMNIST (BG) between vanilla DM and FairDD + DM.

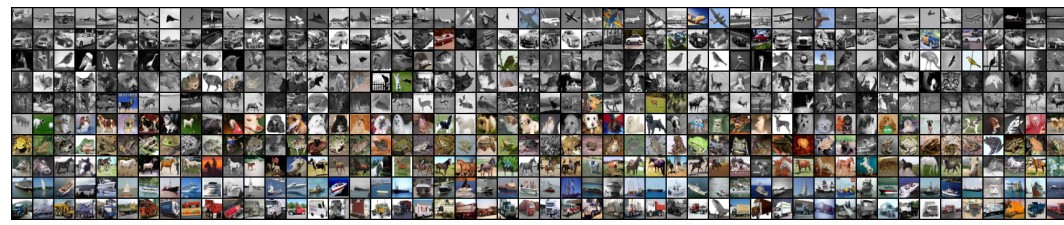

(a) Visualization of the initialized dataset at IPC = 50 in CIFAR10-S. The top five classes (rows) are dominated by the grayscale images, and color ones dominate the bottle five.

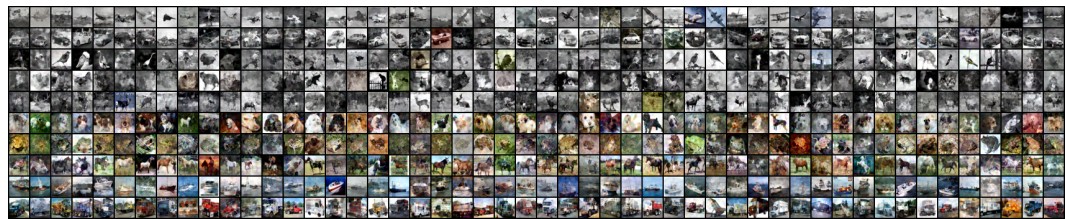

(b) Visualization of the condensed dataset at IPC = 50 in CIFAR10-S using Vanilla DM. The foreground and background of each class inherit the bias.

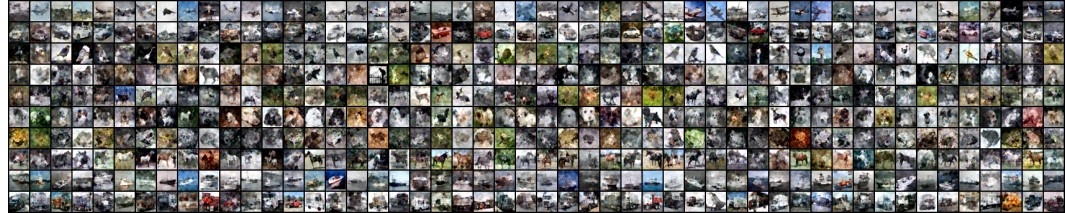

(c) Visualization of the condensed dataset at IPC = 50 in CIFAR10-S using FairDD + DM. The foreground and background of each class mitigate such bias.

Figure 12: Visualization comparison on CIFAR10-S between vanilla DM and FairDD + DM.

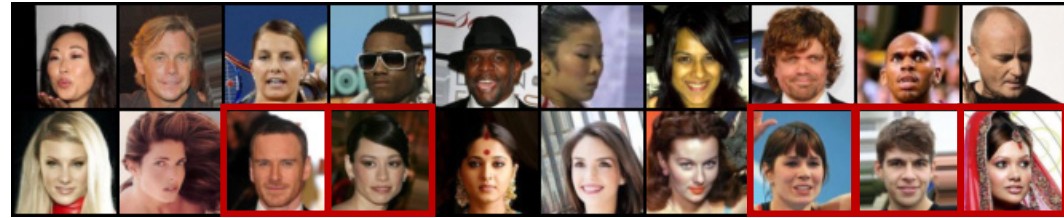

(a) Visualization of the initialized dataset at IPC = 10 in CelebA. The top row is dominated by the male, and the female dominates the bottom row.

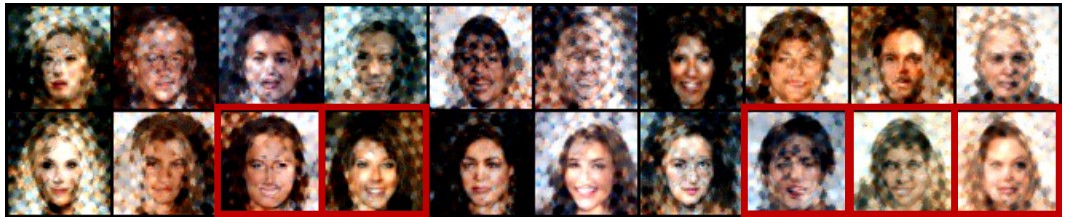

(b) Visualization of the condensed dataset at IPC = 10 in CelebA using Vanilla DM. The synthetic dataset inherits the gender bias.

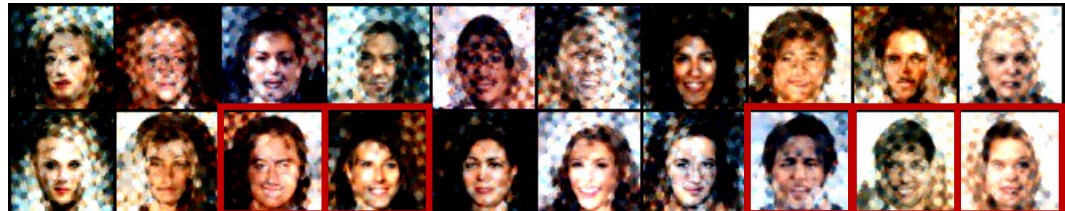

(c) Visualization of the condensed dataset at IPC = 10 in CelebA using FairDD + DM. The synthetic dataset mitigates the gender bias.

Figure 13: FVisualization comparison on CelebA between vanilla DM and FairDD + DM.

# O   COMPLETE RESULTS IN THE FORMAT OF MEAN ± STANDARD DEVIATION.

Table 21: Fairness comparison on diverse IPCs. The results are reported in the format of mean ± standard deviation.

| Methods Dataset | IPC | Random | | DM | | DM+FairDD | | DC | | DC+FairDD | | IDC | | IDC+FairDD | | DREAM | | +FairDD | | Whole | |
|---|---|---|---|---|---|---|---|---|---|---|---|---|---|---|---|---|---|---|---|---|---|
| | | $DEO_M$ | $DEO_A$ | $DEO_M$ | $DEO_A$ | $DEO_M$ | $DEO_A$ | $DEO_M$ | $DEO_A$ | $DEO_M$ | $DEO_A$ | $DEO_M$ | $DEO_A$ | $DEO_M$ | $DEO_A$ | $DEO_M$ | $DEO_A$ | $DEO_M$ | $DEO_A$ | $DEO_M$ | $DEO_A$ |
| C-MNIST (FG) | 10 | 100.0±0.00 | 98.72±0.40 | 100.0±0.00 | 99.96±0.05 | 17.04±1.13 | 7.95±0.59 | 99.85±0.00 | 65.61±2.14 | 26.75±2.78 | 11.96±0.59 | 100.0±0.00 | 91.45±1.10 | 12.24±1.18 | 6.64±0.44 | 98.99±1.96 | 78.71±2.96 | 11.88±0.79 | 7.21±0.39 | 10.10±0.40 | 5.89±0.29 |
| | 50 | 100.0±0.00 | 99.58±0.12 | 100.0±0.00 | 99.68±1.67 | 10.05±0.75 | 5.46±0.40 | 46.99±1.39 | 20.55±1.03 | 18.42±1.39 | 8.86±0.35 | 65.34±2.98 | 34.91±1.40 | 9.18±1.26 | 5.94±0.42 | 52.03±1.80 | 26.63±2.52 | 18.37±1.53 | 7.50±0.13 | | |
| | 100 | 100.0±0.00 | 88.64±2.67 | 99.36±0.27 | 66.38±1.80 | 8.17±0.45 | 4.86±0.21 | 45.27±1.53 | 17.45±0.99 | 22.32±1.85 | 9.49±0.94 | 64.36±1.63 | 35.82±1.59 | 11.88±1.65 | 6.21±0.30 | 69.30±1.87 | 33.30±1.55 | 11.88±1.24 | 6.88±0.34 | | |
| C-MNIST (BG) | 10 | 100.0±0.00 | 99.11±0.26 | 100.0±0.00 | 99.97±0.04 | 13.42±1.61 | 6.77±0.41 | 100.0±0.00 | 73.60±2.31 | 20.66±1.58 | 9.94±0.57 | 100.0±0.00 | 88.30±2.47 | 18.61±1.52 | 7.50±0.26 | 100.0±2.50 | 52.06±2.16 | 15.31±0.66 | 6.83±0.63 | 9.70±0.74 | 5.78±0.11 |
| | 50 | 100.0±0.00 | 99.77±0.09 | 100.0±0.00 | 97.85±1.11 | 8.98±0.74 | 5.25±0.22 | 60.66±3.05 | 26.38±1.59 | 20.29±2.00 | 9.90±0.54 | 93.05±3.35 | 42.23±1.87 | 19.66±2.48 | 8.05±0.31 | 64.15±1.50 | 23.30±2.94 | 20.41±1.22 | 9.04±0.21 | | |
| | 100 | 100.0±0.00 | 89.07±2.30 | 100.0±0.00 | 52.23±2.46 | 6.60±0.45 | 4.31±0.19 | 62.63±1.85 | 20.87±1.28 | 32.58±1.97 | 10.40±0.64 | 63.24±1.37 | 27.79±1.42 | 12.24±1.38 | 6.32±0.53 | 44.88±1.34 | 22.86±2.22 | 16.33±1.12 | 7.80±0.25 | | |
| C-FMNIST (FG) | 10 | 100.0±0.00 | 99.18±0.11 | 100.0±0.00 | 99.05±0.14 | 26.87±1.69 | 16.38±0.99 | 99.40±0.47 | 78.96±1.98 | 46.80±2.98 | 24.01±1.06 | 100.0±0.00 | 97.27±1.44 | 32.33±1.22 | 16.80±1.45 | 100.0±0.00 | 95.17±1.20 | 42.00±2.45 | 20.87±0.87 | 79.20±0.75 | 41.72±0.99 |
| | 50 | 100.0±0.00 | 94.61±1.08 | 100.0±0.00 | 96.46±0.33 | 24.92±1.89 | 13.74±0.68 | 99.33±0.50 | 67.02±1.00 | 46.67±2.43 | 21.48±0.84 | 100.0±0.00 | 81.93±0.81 | 40.00±2.15 | 17.37±0.87 | 99.67±0.22 | 83.27±1.87 | 47.67±0.00 | 22.33±1.20 | | |
| | 100 | 100.0±0.00 | 94.85±0.74 | 100.0±0.00 | 85.11±1.09 | 23.83±1.92 | 12.75±0.41 | 99.58±0.37 | 66.45±2.06 | 56.68±1.96 | 23.07±1.92 | 100.0±0.00 | 79.10±2.04 | 48.33±1.37 | 17.43±0.79 | 97.33±0.40 | 70.10±2.25 | 74.00±1.22 | 40.40±1.00 | | |
| C-FMNIST (BG) | 10 | 100.0±0.00 | 99.40±0.14 | 100.0±0.00 | 99.68±0.05 | 33.05±2.35 | 19.72±1.18 | 100.0±0.00 | 92.91±0.95 | 61.75±2.23 | 34.88±1.16 | 100.0±0.00 | 99.40±0.25 | 42.00±1.59 | 23.80±0.28 | 100.0±0.00 | 94.70±0.76 | 36.00±2.48 | 23.50±0.64 | 91.40±1.85 | 51.68±1.34 |
| | 50 | 100.0±0.00 | 98.52±0.63 | 100.0±0.00 | 99.71±0.10 | 24.50±1.19 | 14.47±0.66 | 100.0±0.00 | 75.41±1.20 | 44.60±1.25 | 25.25±0.99 | 100.0±0.00 | 95.60±1.43 | 78.00±2.87 | 34.50±1.54 | 100.0±0.00 | 88.40±1.30 | 34.00±3.37 | 23.70±1.02 | | |
| | 100 | 100.0±0.00 | 96.05±0.67 | 100.0±0.00 | 93.88±0.66 | 21.95±1.11 | 13.33±0.80 | 99.70±0.37 | 73.38±1.80 | 52.75±3.39 | 23.48±1.60 | 100.0±0.00 | 90.70±1.82 | 77.00±2.58 | 36.00±2.62 | 100.0±0.00 | 83.90±1.56 | 40.00±2.48 | 23.20±0.94 | | |
| CIFAR10-S | 10 | 25.04±0.93 | 8.29±0.65 | 59.20±1.27 | 39.31±1.28 | 31.75±1.52 | 8.73±0.71 | 42.23±1.83 | 27.35±1.52 | 22.08±1.16 | 8.22±1.09 | 80.70±2.83 | 48.38±1.20 | 19.90±2.95 | 5.28±0.87 | 51.80±3.18 | 31.43±0.39 | 20.80±2.26 | 7.77±0.35 | 49.72±0.84 | 33.17±0.32 |
| | 50 | 57.11±1.42 | 28.89±1.83 | 75.13±1.24 | 55.70±1.11 | 18.28±0.78 | 7.35±0.59 | 71.46±1.00 | 45.81±2.23 | 34.39±2.22 | 11.21±0.84 | 92.00±0.61 | 60.56±0.78 | 29.00±1.75 | 9.10±1.04 | 56.80±1.31 | 36.19±0.59 | 14.70±1.29 | 6.53±0.80 | | |
| | 100 | 66.49±1.88 | 43.16±1.45 | 73.81±2.11 | 55.10±1.45 | 14.77±0.50 | 5.89±0.29 | 68.69±1.53 | 48.64±1.17 | 32.70±1.58 | 11.26±0.97 | 92.70±0.92 | 60.93±0.51 | 62.80±2.97 | 25.18±1.61 | 82.30±2.16 | 48.12±2.91 | 12.10±2.71 | 6.06±0.80 | | |
| CelebA | 10 | 10.48±1.97 | 9.20±2.00 | 30.01±1.34 | 28.85±1.33 | 9.37±0.82 | 5.71±0.85 | 15.48±2.62 | 14.16±2.59 | 6.64±1.60 | 5.29±1.62 | 34.85±1.51 | 34.48±1.45 | 8.36±2.37 | 4.49±1.75 | 40.75±1.35 | 36.70±1.31 | 9.20±1.20 | 5.36±0.89 | 24.85±0.90 | 24.16±0.54 |
| | 50 | 22.88±1.41 | 20.32±1.02 | 40.26±0.94 | 38.81±0.70 | 14.08±0.32 | 9.87±1.00 | 24.89±3.32 | 23.83±3.18 | 14.33±1.35 | 12.92±1.30 | 56.74±2.05 | 46.50±1.43 | 22.57±1.37 | 15.15±0.94 | 43.57±1.60 | 38.53±0.69 | 23.62±1.54 | 14.29±0.88 | | |
| | 100 | 18.67±1.19 | 18.01±1.16 | 42.63±1.18 | 41.12±0.96 | 10.93±0.46 | 6.65±0.56 | 29.00±2.42 | 27.52±2.21 | 18.16±2.80 | 17.04±2.93 | 50.99±1.02 | 42.66±0.34 | 28.27±1.65 | 17.63±1.61 | 52.51±0.84 | 39.34±0.50 | 24.87±1.82 | 15.36±1.26 | | |

Table 22: Accuracy comparison on diverse IPCs. The results are reported in the format of mean $\pm$ standard deviation.

| Methods Datasets | IPC | Random Acc. | DM Acc. | +FairDD Acc. | DC Acc. | +FairDD Acc. | IDC Acc. | +FairDD Acc. | DREAM Acc. | +FairDD Acc. | Whole Acc. |
|---|---|---|---|---|---|---|---|---|---|---|---|
| C-MNIST (FG) | 10 | $30.75_{\pm0.96}$ | $25.01_{\pm0.94}$ | $94.61_{\pm0.21}$ | $71.41_{\pm1.27}$ | $90.62_{\pm0.28}$ | $53.06_{\pm1.13}$ | $95.67_{\pm0.20}$ | $75.04_{\pm1.86}$ | $94.04_{\pm0.14}$ | |
| | 50 | $47.38_{\pm0.98}$ | $56.84_{\pm1.92}$ | $96.58_{\pm0.08}$ | $90.54_{\pm0.43}$ | $92.68_{\pm0.18}$ | $88.55_{\pm0.38}$ | $96.77_{\pm0.07}$ | $91.02_{\pm0.68}$ | $94.59_{\pm0.23}$ | $97.71_{\pm0.09}$ |
| | 100 | $67.41_{\pm1.08}$ | $78.04_{\pm1.24}$ | $96.79_{\pm0.14}$ | $91.64_{\pm0.33}$ | $93.23_{\pm0.19}$ | $90.39_{\pm0.48}$ | $97.11_{\pm0.08}$ | $88.87_{\pm1.04}$ | $95.16_{\pm0.12}$ | |
| C-MNIST (BG) | 10 | $27.95_{\pm0.75}$ | $23.40_{\pm0.57}$ | $94.88_{\pm0.13}$ | $65.91_{\pm1.91}$ | $90.84_{\pm0.19}$ | $62.09_{\pm1.06}$ | $94.84_{\pm0.21}$ | $79.81_{\pm1.37}$ | $93.54_{\pm0.26}$ | |
| | 50 | $45.52_{\pm0.98}$ | $47.74_{\pm1.35}$ | $96.86_{\pm0.09}$ | $88.53_{\pm0.61}$ | $92.20_{\pm0.19}$ | $86.14_{\pm0.74}$ | $95.29_{\pm0.13}$ | $89.24_{\pm0.82}$ | $93.20_{\pm0.50}$ | $97.80_{\pm0.07}$ |
| | 100 | $67.28_{\pm1.31}$ | $79.87_{\pm0.77}$ | $97.33_{\pm0.09}$ | $90.20_{\pm0.57}$ | $92.73_{\pm0.17}$ | $89.66_{\pm0.66}$ | $95.84_{\pm0.06}$ | $90.70_{\pm0.70}$ | $94.06_{\pm0.24}$ | |
| C-FMNIST (FG) | 10 | $32.80_{\pm1.44}$ | $33.35_{\pm1.27}$ | $77.09_{\pm0.33}$ | $60.77_{\pm0.88}$ | $76.01_{\pm0.19}$ | $44.08_{\pm1.64}$ | $79.66_{\pm0.21}$ | $49.72_{\pm1.07}$ | $77.24_{\pm0.15}$ | |
| | 50 | $42.48_{\pm1.05}$ | $49.94_{\pm0.75}$ | $82.11_{\pm0.20}$ | $69.08_{\pm0.54}$ | $75.83_{\pm0.33}$ | $64.45_{\pm0.69}$ | $80.80_{\pm0.33}$ | $65.69_{\pm1.04}$ | $78.79_{\pm1.07}$ | $82.94_{\pm0.14}$ |
| | 100 | $55.31_{\pm0.67}$ | $57.99_{\pm0.84}$ | $83.25_{\pm0.19}$ | $68.84_{\pm0.61}$ | $74.91_{\pm0.40}$ | $66.37_{\pm0.26}$ | $80.28_{\pm0.19}$ | $68.25_{\pm0.94}$ | $78.51_{\pm0.55}$ | |
| C-FMNIST (BG) | 10 | $24.96_{\pm0.63}$ | $22.26_{\pm0.77}$ | $71.10_{\pm0.43}$ | $47.32_{\pm0.96}$ | $68.51_{\pm0.38}$ | $37.59_{\pm0.76}$ | $72.67_{\pm0.20}$ | $45.30_{\pm0.78}$ | $71.56_{\pm0.34}$ | |
| | 50 | $34.92_{\pm0.57}$ | $36.27_{\pm0.72}$ | $79.07_{\pm0.27}$ | $60.58_{\pm0.72}$ | $75.80_{\pm0.28}$ | $46.20_{\pm1.10}$ | $73.72_{\pm0.43}$ | $53.62_{\pm0.50}$ | $72.80_{\pm0.11}$ | $77.97_{\pm0.44}$ |
| | 100 | $44.87_{\pm0.65}$ | $49.30_{\pm0.58}$ | $80.63_{\pm0.16}$ | $62.70_{\pm0.66}$ | $71.76_{\pm0.28}$ | $48.61_{\pm1.00}$ | $73.18_{\pm0.57}$ | $53.32_{\pm0.78}$ | $73.00_{\pm0.39}$ | |
| CIFAR10-S | 10 | $23.60_{\pm0.32}$ | $37.88_{\pm0.27}$ | $45.17_{\pm0.46}$ | $37.88_{\pm0.76}$ | $41.82_{\pm0.70}$ | $48.30_{\pm0.79}$ | $56.40_{\pm0.37}$ | $55.09_{\pm0.43}$ | $58.40_{\pm0.24}$ | |
| | 50 | $36.46_{\pm0.49}$ | $45.02_{\pm0.44}$ | $58.84_{\pm0.23}$ | $41.28_{\pm0.80}$ | $49.26_{\pm0.42}$ | $47.26_{\pm0.73}$ | $57.84_{\pm0.24}$ | $57.59_{\pm0.93}$ | $61.85_{\pm0.33}$ | $69.78_{\pm0.18}$ |
| | 100 | $39.34_{\pm0.56}$ | $48.11_{\pm0.63}$ | $61.33_{\pm0.37}$ | $42.73_{\pm0.73}$ | $51.74_{\pm0.52}$ | $47.27_{\pm1.66}$ | $56.98_{\pm0.85}$ | $57.14_{\pm0.26}$ | $62.70_{\pm0.37}$ | |
| CelebA | 10 | $54.51_{\pm1.63}$ | $61.79_{\pm0.82}$ | $64.37_{\pm0.31}$ | $57.19_{\pm2.31}$ | $57.63_{\pm1.49}$ | $61.49_{\pm0.57}$ | $63.54_{\pm0.73}$ | $64.38_{\pm0.33}$ | $66.26_{\pm0.18}$ | |
| | 50 | $55.99_{\pm1.23}$ | $64.61_{\pm0.25}$ | $68.50_{\pm0.58}$ | $60.16_{\pm1.57}$ | $59.89_{\pm1.11}$ | $60.75_{\pm0.37}$ | $66.89_{\pm0.28}$ | $64.62_{\pm0.44}$ | $68.26_{\pm0.37}$ | $74.09_{\pm0.27}$ |
| | 100 | $60.62_{\pm0.71}$ | $65.13_{\pm0.30}$ | $68.84_{\pm0.32}$ | $62.53_{\pm2.42}$ | $61.89_{\pm1.91}$ | $64.04_{\pm0.68}$ | $67.24_{\pm0.58}$ | $62.58_{\pm0.45}$ | $64.12_{\pm0.28}$ | |

Table 23: Cross-arch. comparison. The results are reported in the format of mean $\pm$ standard deviation.

| Method | Cross arch. | DM $\overline{\text{DEO}_M}$ | $\overline{\text{DEO}_A}$ | $\overline{\text{Acc.}}$ | DM+FairDD $\overline{\text{DEO}_M}$ | $\overline{\text{DEO}_A}$ | $\overline{\text{Acc.}}$ |
|---|---|---|---|---|---|---|---|
| C-MNIST (FG) | ConvNet | $100.0_{\pm 0.00}$ | $91.68_{\pm 1.67}$ | $56.84_{\pm 1.92}$ | $10.05_{\pm 0.75}$ | $5.46_{\pm 0.40}$ | $96.58_{\pm 0.80}$ |
| | AlexNet | $100.0_{\pm 0.00}$ | $98.82_{\pm 1.63}$ | $44.02_{\pm 3.34}$ | $10.35_{\pm 0.60}$ | $6.16_{\pm 0.39}$ | $96.12_{\pm 0.19}$ |
| | VGG11 | $99.70_{\pm 0.02}$ | $70.73_{\pm 3.44}$ | $75.22_{\pm 1.41}$ | $9.55_{\pm 0.66}$ | $5.39_{\pm 0.36}$ | $96.80_{\pm 0.16}$ |
| | ResNet18 | $100.0_{\pm 0.00}$ | $96.00_{\pm 1.07}$ | $52.05_{\pm 1.93}$ | $8.40_{\pm 0.50}$ | $4.63_{\pm 0.27}$ | $97.13_{\pm 0.18}$ |
| | Mean | 99.93 | 89.31 | 57.03 | 9.59 | 5.41 | 96.66 |
| C-FMNIST (BG) | ConvNet | $100.0_{\pm 0.00}$ | $99.71_{\pm 0.10}$ | $36.27_{\pm 0.72}$ | $24.50_{\pm 1.19}$ | $14.47_{\pm 0.66}$ | $79.07_{\pm 0.27}$ |
| | AlexNet | $100.0_{\pm 0.00}$ | $99.75_{\pm 0.12}$ | $22.72_{\pm 1.60}$ | $20.60_{\pm 1.03}$ | $14.11_{\pm 0.85}$ | $76.14_{\pm 0.71}$ |
| | VGG11 | $100.0_{\pm 0.00}$ | $97.77_{\pm 0.98}$ | $43.11_{\pm 0.79}$ | $21.60_{\pm 0.76}$ | $14.36_{\pm 0.91}$ | $78.57_{\pm 0.24}$ |
| | ResNet18 | $100.0_{\pm 0.00}$ | $99.78_{\pm 0.09}$ | $23.37_{\pm 1.11}$ | $22.50_{\pm 1.09}$ | $14.96_{\pm 0.81}$ | $75.21_{\pm 0.53}$ |
| | Mean | 100.0 | 99.25 | 31.37 | 22.30 | 14.73 | 77.25 |
| CIFAR10-S | ConvNet | $75.13_{\pm 1.24}$ | $55.70_{\pm 1.11}$ | $45.02_{\pm 0.44}$ | $18.28_{\pm 0.78}$ | $7.35_{\pm 0.59}$ | $58.84_{\pm 0.23}$ |
| | AlexNet | $75.30_{\pm 0.90}$ | $52.57_{\pm 1.29}$ | $36.09_{\pm 0.51}$ | $15.84_{\pm 1.11}$ | $5.12_{\pm 0.28}$ | $49.16_{\pm 0.66}$ |
| | VGG11 | $61.48_{\pm 1.91}$ | $44.05_{\pm 1.40}$ | $43.23_{\pm 0.55}$ | $11.51_{\pm 0.77}$ | $4.16_{\pm 0.32}$ | $52.65_{\pm 0.76}$ |
| | ResNet18 | $76.23_{\pm 0.72}$ | $54.35_{\pm 0.99}$ | $38.03_{\pm 0.51}$ | $16.44_{\pm 0.98}$ | $5.14_{\pm 0.50}$ | $50.93_{\pm 0.77}$ |
| | Mean | 72.04 | 51.67 | 40.59 | 15.27 | 5.44 | 52.90 |
| CelebA | ConvNet | $40.26_{\pm 0.94}$ | $38.81_{\pm 0.70}$ | $64.61_{\pm 0.25}$ | $14.08_{\pm 0.32}$ | $9.87_{\pm 1.00}$ | $68.50_{\pm 0.58}$ |
| | AlexNet | $32.51_{\pm 0.85}$ | $31.62_{\pm 1.03}$ | $63.10_{\pm 0.55}$ | $9.38_{\pm 0.86}$ | $5.75_{\pm 0.76}$ | $64.24_{\pm 0.80}$ |
| | VGG11 | $26.03_{\pm 2.10}$ | $24.63_{\pm 1.97}$ | $61.57_{\pm 0.79}$ | $8.95_{\pm 0.97}$ | $6.32_{\pm 1.10}$ | $62.05_{\pm 1.61}$ |
| | ResNet18 | $25.60_{\pm 1.87}$ | $24.93_{\pm 1.75}$ | $60.32_{\pm 0.82}$ | $6.72_{\pm 0.81}$ | $4.29_{\pm 0.67}$ | $61.80_{\pm 1.07}$ |
| | Mean | 31.10 | 30.25 | 62.40 | 9.78 | 6.58 | 64.15 |

Table 24: Ablation on BR at IPC = 50. The results are reported in the format of mean $\pm$ standard deviation.

| Methods Dataset | BR | DM $\overline{\text{DEO}_M}$ | $\overline{\text{DEO}_A}$ | $\overline{\text{Acc.}}$ | DM+FairDD $\overline{\text{DEO}_M}$ | $\overline{\text{DEO}_A}$ | $\overline{\text{Acc.}}$ |
|---|---|---|---|---|---|---|---|
| C-MNIST (FG) | 0.85 | $99.54_{\pm 0.37}$ | $70.13_{\pm 1.69}$ | $76.24_{\pm 1.68}$ | $10.13_{\pm 0.75}$ | $5.20_{\pm 0.33}$ | $96.62_{\pm 0.11}$ |
| | 0.90 | $100.0_{\pm 0.00}$ | $91.68_{\pm 1.67}$ | $56.84_{\pm 1.92}$ | $10.05_{\pm 0.75}$ | $5.46_{\pm 0.40}$ | $96.58_{\pm 0.08}$ |
| | 0.95 | $100.0_{\pm 0.00}$ | $100.0_{\pm 0.00}$ | $33.73_{\pm 1.08}$ | $10.30_{\pm 0.74}$ | $5.84_{\pm 0.39}$ | $96.05_{\pm 0.17}$ |
| C-FMNIST (BG) | 0.85 | $100.0_{\pm 0.00}$ | $95.54_{\pm 0.79}$ | $46.14_{\pm 0.93}$ | $23.75_{\pm 1.58}$ | $13.85_{\pm 0.82}$ | $79.61_{\pm 0.17}$ |
| | 0.90 | $100.0_{\pm 0.00}$ | $99.71_{\pm 0.10}$ | $36.27_{\pm 0.72}$ | $24.50_{\pm 1.19}$ | $14.47_{\pm 0.66}$ | $79.07_{\pm 0.27}$ |
| | 0.95 | $100.0_{\pm 0.00}$ | $99.79_{\pm 0.08}$ | $26.30_{\pm 0.44}$ | $29.15_{\pm 1.30}$ | $17.72_{\pm 0.90}$ | $78.46_{\pm 0.25}$ |
| CIFAR10-S | 0.85 | $71.75_{\pm 0.90}$ | $50.11_{\pm 0.70}$ | $46.99_{\pm 0.43}$ | $16.44_{\pm 0.88}$ | $6.58_{\pm 0.76}$ | $59.12_{\pm 0.32}$ |
| | 0.90 | $75.13_{\pm 1.24}$ | $55.70_{\pm 1.11}$ | $45.02_{\pm 0.44}$ | $18.28_{\pm 0.78}$ | $7.35_{\pm 0.59}$ | $58.84_{\pm 0.23}$ |
| | 0.95 | $75.43_{\pm 1.28}$ | $58.58_{\pm 0.82}$ | $43.56_{\pm 0.38}$ | $17.49_{\pm 1.26}$ | $7.10_{\pm 0.91}$ | $58.18_{\pm 0.30}$ |

Table 25: Ablation on initialization at IPC = 50. The results are reported in the format of mean $\pm$ standard deviation.

| Methods Dataset | Init. | DM | | | DM+FairDD | | |
|---|---|---|---|---|---|---|---|
| | | $\overline{DEO_M}$ | $\overline{DEO_A}$ | $\overline{Acc.}$ | $\overline{DEO_M}$ | $\overline{DEO_A}$ | $\overline{Acc.}$ |
| C-MNIST (FG) | Random | $100.0_{\pm0.00}$ | $91.68_{\pm1.67}$ | $56.84_{\pm1.92}$ | $10.05_{\pm0.75}$ | $5.46_{\pm0.40}$ | $96.58_{\pm0.08}$ |
| | Noise | $100.0_{\pm0.00}$ | $99.64_{\pm0.39}$ | $41.45_{\pm1.46}$ | $9.33_{\pm0.63}$ | $5.28_{\pm0.29}$ | $96.06_{\pm0.10}$ |
| | Hybrid | $100.0_{\pm0.00}$ | $99.06_{\pm0.74}$ | $39.97_{\pm1.77}$ | $9.03_{\pm0.62}$ | $5.33_{\pm0.25}$ | $96.27_{\pm0.09}$ |
| C-FMNIST (BG) | Random | $100.0_{\pm0.00}$ | $99.71_{\pm0.10}$ | $36.27_{\pm0.72}$ | $24.50_{\pm1.19}$ | $14.47_{\pm0.66}$ | $79.07_{\pm0.27}$ |
| | Noise | $100.0_{\pm0.00}$ | $99.67_{\pm0.07}$ | $22.92_{\pm0.78}$ | $23.00_{\pm1.93}$ | $14.40_{\pm0.63}$ | $78.84_{\pm0.20}$ |
| | Hybrid | $100.0_{\pm0.00}$ | $99.68_{\pm0.07}$ | $26.38_{\pm0.80}$ | $21.45_{\pm1.32}$ | $14.34_{\pm0.89}$ | $79.19_{\pm0.16}$ |
| CIFAR10-S | Random | $75.13_{\pm1.24}$ | $55.70_{\pm1.11}$ | $45.02_{\pm0.44}$ | $18.28_{\pm0.78}$ | $7.35_{\pm0.59}$ | $58.84_{\pm0.23}$ |
| | Noise | $55.28_{\pm1.57}$ | $37.26_{\pm0.63}$ | $46.97_{\pm0.29}$ | $16.15_{\pm0.62}$ | $6.13_{\pm0.50}$ | $56.41_{\pm0.30}$ |
| | Hybrid | $65.59_{\pm1.38}$ | $46.18_{\pm0.84}$ | $45.16_{\pm0.40}$ | $17.30_{\pm1.26}$ | $6.71_{\pm0.33}$ | $56.78_{\pm0.24}$ |

