# OpenReview forum: "FairDD: Fair Dataset Distillation via Adversarial Matching"
_ICLR.cc/2025/Conference — Submitted to ICLR 2025_

### Official Review · Reviewer_SjG8 · 2024-11-02

**Soundness:** 3
**Presentation:** 3
**Contribution:** 3
**Rating:** 6
**Confidence:** 4

**Summary:**

This paper addresses attribute imbalance in dataset distillation, focusing on improving the fairness of condensed datasets. The authors present a unified perspective on data matching and identify unfairness issues in conventional distillation methods regarding protected attributes. To address this, they introduce an adversarial matching loss function that ensures equal contribution from different attribute groups. Their theoretical analysis demonstrates both the equitable treatment of attribute groups and the preservation of vanilla data matching optimization. Experimental results on C-MNIST, C-FMNIST, CIFAR10-S, and CelebA datasets provide both quantitative and qualitative evidence of the method's effectiveness in mitigating unfairness compared to naive matching approaches.

**Strengths:**

1. This is the first work to identify attribute unfairness in current data distillation methods.

2. The authors propose a simple yet effective method to solve this unfairness.

3. The authors provide theoretical justification for their proposed method in addressing the unfairness issue.

4. The authors present comprehensive experiments to demonstrate the soundness of their proposed method.

5. The paper is well-written and easy to follow.

**Weaknesses:**

1. Including Vision Transformer architectures as backbone networks would further demonstrate the method's generalizability.

2. Examining its performance on more challenging datasets like CIFAR-100 or ImageNet would strengthen its practical applicability.

3. Including visual examples from the CelebA dataset in the supplementary material would help readers better understand the fairness improvements achieved by the proposed method.

4. The term "adversarial matching" could benefit from additional clarification, as the current mathematical formulation doesn't explicitly show adversarial operations. A brief explanation of this terminology would enhance the paper's clarity.

**Questions:**

Please see the weaknesses

---

> ### Author Response · Authors · 2024-11-22
> **Response to Reviewer SjG8**
>
> **W1: Including Vision Transformer architectures as backbone networks would further demonstrate the method's generalizability.**
>
> | Methods (Dataset)   |IPC || DM ||| DM+FairDD ||
> |----------------------|-----|:------------:|:------------:|:------------:|:------------:|:------------:|:------------:|
> |    | | **Acc.**| **$DEO_M$**| **$DEO_A$**| **Acc.**| **$DEO_M$**| **$DEO_A$**|
> | ViT1| 10  | 18.63|100.0|98.48|56.15|82.10|56.72|
> | ViT2 | 10  | 18.28|100.0|98.99|33.89|72.85|40.97|
> | ViT3 | 10  | 16.15|100.0|95.75|26.70|65.71|29.46|
>
> Thank you for your comments. Although the Vision Transformer (ViT) is a powerful backbone network, to the best of my knowledge, current DDs, such as DM and DC, have not yet utilized ViT as the extraction network.
>
> We conducted experiments using 1-layer, 2-layer, and 3-layer ViTs. As shown in Table, vanilla DM at IPC=10 suffers performance degradation in classification, dropping from 25.01\% to 18.63\%. Moreover, as the number of layers increases, the performance deteriorates more severely. This suggests that current DDs are not directly compatible with ViTs.
>
> While FairDD still outperforms DM in both accuracy and fairness metrics, the observed improvement gain is smaller compared to results obtained on convolutional networks. Further research into leveraging ViTs for DD and FairDD is a promising direction worth exploring.
>
> **W2: Examining its performance on more challenging datasets like CIFAR100 or ImageNet would strengthen its practical applicability.**
>
> | Methods (Dataset)   | IPC || Whole ||| DM || | DM+FairDD ||
> |----------------------|-----|:------------:|:------------:|:------------:|:------------:|:------------:|:------------:|:------------:|:------------:|:------------:|
> |    | | **Acc.**| **$DEO_M$**| **$DEO_A$**| **Acc.**| **$DEO_M$**| **$DEO_A$**| **Acc.**| **$DEO_M$**| **$DEO_A$**|
> | CIFAR100-S  | 10  | 38.98|65.60|31.21|19.69|69.90|25.37|22.84|42.60|9.83|
>
> Thanks for your suggestions. We created CIFAR100-S following the same operation as CIFAR10-S, where the grayscale or not is regarded as PA. Due to the time limit, we supplemented CIFAR100-S on DM at IPC=10. DM achieves the classification accuracy of 22.84\%, and the fairness of 69.9\% DEO$_M$ and 25.37\% DEO$_A$. Compared to vanilla DM, FairDD obtains more accurate classification performance and mitigates the bias to the minority groups, with 27.30\% DEO$_M$ and 15.54\% DEO$_A$ improvement.
>
> **W3: Including visual examples from the CelebA dataset in the supplementary material would help readers better understand the fairness improvements achieved by the proposed method.**
>
> Thank you for your insightful comments. We have supplemented the visualizations in `Figure 13` in the Appendix of the revised version. The target attribute is attractive, and the protected attribute is gender. The top subfigure shows the initialized synthetic dataset, where the first row is dominated by males, and the second row is dominated by females. The middle subfigure displays the synthetic dataset generated by vanilla DM, which inherits the gender bias. In comparison, the images highlighted by red circles in the last subfigure are transferred from females (majority) to males (minority). This indicates that FairDD effectively mitigates bias by balancing the sample number across PA groups.
>
> **W4: The term "adversarial matching" could benefit from additional clarification, as the current mathematical formulation doesn't explicitly show adversarial operations. A brief explanation of this terminology would enhance the paper's clarity.**
>
> Thanks for your suggestion. We have supplemented the illustration at line 199 in the revised version:
> `Compared to vanilla DDs, which simply pull the synthetic dataset toward the whole dataset center that is biased toward the majority group in the synthetic dataset, FairDD proposes a group-level adversarial alignment, in which each group attracts the synthetic data toward itself, thus forcing it to move farther from other groups. This "pull-and-push" process prevents the synthetic dataset from collapsing into majority groups (fairness) and ensures its class-level distributional coverage (accuracy). `

---

> > ### Comment · Reviewer_SjG8 · 2024-11-26
> > **Thanks for the authors' responses**
> >
> > Thank you for the authors' response. After reviewing it, I find all my concerns resolved. Thus, I will maintain my score in support of accepting this paper.

---

> > > ### Author Response · Authors · 2024-11-26
> > >
> > > We are happy to hear that your concerns have been addressed. Thank you for acknowledging our work. Your encouragement means a lot to us.

---

### Official Review · Reviewer_j9J1 · 2024-11-03

**Soundness:** 3
**Presentation:** 3
**Contribution:** 3
**Rating:** 8
**Confidence:** 3

**Summary:**

This paper addresses a critical issue in dataset distillation (DD), where biases inherent in original datasets tend to amplify within condensed datasets, often exacerbating unfairness toward minority groups. The authors propose "FairDD," a fair dataset distillation framework designed to mitigate these biases by integrating adversarial matching for protected attribute (PA)-wise groups, such as gender and race. Unlike conventional DD approaches that indiscriminately align to the entire dataset distribution (often skewed by majority groups), FairDD aligns synthetic datasets with specific PA groups, aiming to prevent dominance by majority representations. This targeted approach allows for more balanced synthetic data generation, maintaining classification accuracy while improving fairness. The paper’s theoretical analyses and extensive experiments show that FairDD outperforms traditional DD methods in fairness metrics without compromising on accuracy.

**Strengths:**

Novel Approach to Fairness in Dataset Distillation: The concept of adversarially aligning synthetic datasets with PA-wise groups is an innovative approach that tackles a commonly overlooked issue in DD. FairDD’s adversarial matching mechanism, focusing on each PA group rather than the overall distribution, is a thoughtful solution that could inspire further research in fair dataset synthesis.

Versatility Across Matching-Based DD Methods: The proposed FairDD framework is adaptable to various matching-based DD methods, as shown by its successful application in both Distribution and Gradient Matching methods. This versatility highlights FairDD’s potential as a generally applicable solution in the DD field.

Extensive Theoretical and Experimental Validation: The authors conducted thorough theoretical analyses to support their framework, providing a solid foundation for understanding why FairDD effectively reduces biases in dataset distillation. Additionally, they validated their approach through extensive experiments, consistently showing improved fairness across different DD methods without sacrificing model accuracy. This combination of theoretical and empirical rigor adds credibility to FairDD's effectiveness and reliability.

Maintaining Accuracy while Enhancing Fairness: An important advantage of FairDD is that it achieves fairness without compromising the target classification performance, addressing a common trade-off in fairness-oriented methods.

**Weaknesses:**

Limited Practicality Discussion: While FairDD’s focus on fairness is commendable, the framework’s real-world applicability could be affected by computational demands introduced by adversarial matching. The authors could discuss the added computational overhead and resource requirements, especially when scaling to larger datasets.

Scalability to Other Protected Attributes: The paper primarily discusses FairDD’s effectiveness concerning attributes like gender and race. However, it is unclear how well this method generalizes to other protected attributes or more nuanced groups within a PA, especially when there are multiple attributes with intersecting biases. An exploration of such scenarios would enhance the robustness of the approach.

Potential Dependency on Original Dataset Quality: The framework assumes that PA groups in the original dataset are balanced enough to train FairDD effectively. In real-world applications, where some PA groups might be underrepresented, this assumption could limit FairDD’s effectiveness. The authors could address how FairDD performs under different levels of dataset imbalance.

Theoretical Justification: While the paper provides theoretical analysis, a deeper exploration of why adversarial matching effectively reduces bias in condensed datasets could strengthen the contribution. Further theoretical insights could add rigor and help clarify the underlying mechanisms driving FairDD’s success.

**Questions:**

Good paper. I have no further questions

---

> ### Author Response · Authors · 2024-11-22
> **Response to Reviewer j9J1 (Part 1)**
>
> **W1: Limited Practicality Discussion: While FairDD’s focus on fairness is commendable, the framework’s real-world applicability could be affected by computational demands introduced by adversarial matching. The authors could discuss the added computational overhead and resource requirements, especially when scaling to larger datasets.**
>
> Thank you for pointing out your concerns. To begin with, we want to clarify that current DDs in DMF complete their training once the total iteration number is reached. For consistency, our experiments use the same hyperparameters, including the total iteration number and batch size. Consequently, the training time in our experiments is independent of the dataset scale.
>
> Actually, we provided the analysis of computational overhead compared to the vanilla methods in Appendix C of the submitted version. There, we evaluate the impact of the number of groups on training time (min) and peak GPU memory consumption (MB) because FairDD performs fine-grained alignment at the group level.
>
> Here, we further supplement the overhead analysis with respect to image resolutions. We conduct experiments on CMNIST, CelebA (32), CelebA (64), and CelebA (96) on DM and DC at IPC=10. DM and DC align different signals, which would bring different effects.
>
> As illustrated in Table, it can be observed that FairDD + DM does not require additional GPU memory consumption but does necessitate more time. The time gap increases from 0.42 minutes to 1.79 minutes as input resolution varies (e.g., CelebA 32 × 32, CelebA 64 × 64, and CelebA 96 × 96); however, the gap remains small. This can be attributed to FairDD performing group-level alignment on features, which is less influenced by input resolution. Notably, although CMNIST and CelebA (32 × 32) share the same resolution, the time gap is more pronounced for CMNIST (e.g., 3 minutes). This is attributed to CMNIST having 10 attributes, whereas CelebA (32 × 32) has only 2 attributes. These indicate that FairDD + DM requires no additional GPU memory consumption. Its additional time depends on both input resolution and the number of groups, but the number of groups more significantly influences it.
>
> As for DC, FairDD requires additional GPU memory and time. Since FairDD + DC explicitly computes group-level gradients, the resulting gradient caches cause FairDD + DC to consume more memory. The small additional consumption is acceptable given the large performance gains in fairness. Additionally, the time gap is relatively larger than that observed between DM and FairDD + DM. Similar to DM, the group number is the primary factor contributing to additional time consumption compared to input resolution.
>
> | Methods (Dataset)   |Group number|| DM || DM+FairDD || DC|| DC+FairDD ||
> |----------------------|-----|:------------:|:------------:|:------------:|:------------:|:------------:|:------------:|:------------:|:------------:|:------------:|
> |    | | **Time**| **Memory**| **Time**| **Memory**| **Time**| **Memory**| **Time**| **Memory**|
> | CMNIST(32)| 10 |15.55min|1227MB|18.55min|1227MB|58.75min|1767MB|83.13min|1893MB|
> | CelebA (32)| 2 | 10.93min|2293MB|11.35min|2293MB|32.98min|2413MB|34.65min|2479MB|
> | CelebA (64)| 2 | 11.18min|8179MB|12.20min|8177MB|43.67min|8525MB|47.07min|8841MB|
> | CelebA (96)| 2 | 12.83min|17975MB|14.62min|17975MB|82.37min|18855MB|86.88min|19437MB|

---

> ### Author Response · Authors · 2024-11-22
> **Response to Reviewer j9J1 (Part 2)**
>
> **W2: Scalability to Other Protected Attributes: The paper primarily discusses FairDD’s effectiveness concerning attributes like gender and race. However, it is unclear how well this method generalizes to other protected attributes or more nuanced groups within a PA, especially when there are multiple attributes with intersecting biases. An exploration of such scenarios would enhance the robustness of the approach.**
>
> Thanks for your insightful comments. We agree that testing more PAs could further help demonstrate the generalization of our method. We regard `blond hair` as the protected attribute and `attractive` as the target attribute, resulting in CelebA$_h$. As illustrated in Table, FairDD+DM obtains 7.76\% $DEO_M$ and 6.02\% $DEO_A$, outperforming DM by 9.25\% and 3.54\%. Accuracy has also been improved.
>
> | Methods (Dataset)   | IPC || Whole ||| DM || | DM+FairDD ||
> |----------------------|-----|:------------:|:------------:|:------------:|:------------:|:------------:|:------------:|:------------:|:------------:|:------------:|
> |    | | **Acc.**| **$DEO_M$**| **$DEO_A$**| **Acc.**| **$DEO_M$**| **$DEO_A$**| **Acc.**| **$DEO_M$**| **$DEO_A$**|
> | CelebA$_h$       | 10  | 75.33|15.53|11.56|63.64|17.01|9.56|64.86|7.76|6.02|
>
> For more nuanced groups, we perform a fine-grained PA division. For example, we consider `gender` and `wearing-necktie` as two correlated attributes and divide them into four groups: `males with a necktie`, `males without a necktie`, `females with a necktie`, and `females without a necktie` (CelebA$_ {g \ n}$). Similarly, we consider `gender` and `paleskin`, and divide them into four groups (CelebA$_ {g \ p}$). Their target attribute is attractive. As shown in Table, FairDD outperforms vanilla DD in the accuracy and fairness performance on these two experiments. The performance on CelebA$_ {g \ n}$ is improved from 57.50\% to 25.00\% on DEO$_M$ and 52.79\% to 21.73\% on DEO$_A$. Accuracy is also improved from 63.25\% to 67.98\%. Similar results can be observed for gender and paleskin. Hence, FairDD can mitigate more fine-grained attribute bias, even when there is an intersection between attributes.
>
> | Methods (Dataset)   |IPC || DM ||| DM+FairDD ||
> |----------------------|-----|:------------:|:------------:|:------------:|:------------:|:------------:|:------------:|
> |    | | **Acc.**| **$DEO_M$**| **$DEO_A$**| **Acc.**| **$DEO_M$**| **$DEO_A$**|
> | CelebA$_{g \ n}$      | 10  | 63.25|57.50|52.79|67.98|25.00|21.73|
> | CelebA$_{g \ p}$      | 10  | 62.48|44.81|41.60|64.37|26.92|19.33|
>
> **W3: Potential Dependency on Original Dataset Quality: The framework assumes that PA groups in the original dataset are balanced enough to train FairDD effectively. In real-world applications, where some PA groups might be underrepresented, this assumption could limit FairDD’s effectiveness. The authors could address how FairDD performs under different levels of dataset imbalance.**
>
> | Methods (Dataset)   |IPC || DM ||| DM+FairDD ||
> |----------------------|-----|:------------:|:------------:|:------------:|:------------:|:------------:|:------------:|
> |    | | **Acc.**| **$DEO_M$**| **$DEO_A$**| **Acc.**| **$DEO_M$**| **$DEO_A$**|
> | CMNIST| 10  | 25.01|100.0|99.96|94.61|17.04|7.95|
> | CMNIST$_{unbalance}$ | 10  | 23.38|100.0|99.89|94.45|16.33|9.01
>
> Thank you for your insightful comments. Your suggestions have inspired us to further study the effect under more biased scenarios. Specifically, we keep the sample number of the majority group in each class invariant and allocate the sample size to the remaining 9 minority groups with increasing ratios, i.e., 1:2:3:4:5:6:7:8:9. We denote this variant CMNIST$_ {unbalance}$ This could help create varying extents of underrepresented samples for different minority groups. Notably, the least-represented PA groups account for only about 1/500 of the entire dataset, which equates to just 12 samples out of 6000 in CMNIST$_ {unbalance}$. As shown in Table 3, FairDD achieves a robust performance of 16.33\% DEO$_M$ and 9.01\% DEO$_A$ compared to 17.04\% and 7.95\% in the balanced PA groups. A similar steady behavior is observed in accuracy, which changes from 94.45\% to 94.61\%. This illustrates the robustness of FairDD under different levels
> of dataset imbalance.

---

> > ### Author Response · Authors · 2024-11-22
> > **Response to Reviewer j9J1 (Part 3)**
> >
> > **W4: Theoretical Justification: While the paper provides theoretical analysis, a deeper exploration of why adversarial matching effectively reduces bias in condensed datasets could strengthen the contribution. Further theoretical insights could add rigor and help clarify the underlying mechanisms driving FairDD’s success.**
> >
> > Thanks for your insightful comments. We are encouraged by your acknowledgment of our theoretical analysis. We provide Theorem 4.1 to illustrate the equal contribution of each PA to the final synthetic datasets for fairness. Theorem 4.2 is given to demonstrate the class-level distributional coverage for classification accuracy. Actually, we have consistently sought to provide more in-depth theoretical analysis to support our experimental results since drafting this version. However, delivering a deeper exploration is challenging, particularly within such a limited timeframe.

---

> > > ### Author Response · Authors · 2024-11-26
> > >
> > > Many thanks for your support of our work. As the discussion period deadline is approaching, please do not hesitate to let us know if you have any further questions. We are more than happy to assist with any remaining concerns.

---

> > > > ### Comment · Reviewer_j9J1 · 2024-11-26
> > > >
> > > > Thank you for the detailed reply. I don't have any further questions. I'll maintain my original score.

---

> > > > > ### Author Response · Authors · 2024-11-27
> > > > >
> > > > > Thank you for acknowledging our work. We’re glad to hear that our responses have addressed your concerns.

---

### Official Review · Reviewer_PZx9 · 2024-11-03

**Soundness:** 3
**Presentation:** 4
**Contribution:** 3
**Rating:** 6
**Confidence:** 4

**Summary:**

The paper addresses a crucial issue: fairness in dataset distillation. The first part of the paper shows that if the original dataset is biased, the distilled dataset exacerbates such biases by generating images primarily from the majority group. Thereafter, the paper proposes a simple modification to the loss function to ensure representation from the different groups in the condensed dataset. The authors have shown theoretical evidence for the efficacy of their method and demonstrated its effectiveness across multiple datasets of varying versions. Overall, this is an issue that demands more research and requires further analyses to ensure that the condensed dataset is fair.

**Strengths:**

1. The loss function FairDD seems intuitive. Instead of optimizing across all samples of a given class, the authors have ensured that each group in the training set gets a fair chance in pulling the condensed dataset towards itself.
2. The analysis on the exacerbation of biases in the distilled dataset is a strong evidence towards the necessity behind further research on this issue.
3. The paper has provided theoretical analysis for their proposed approach.
4. The authors have performed their analysis on a variety of datasets and across multiple model architectures.

**Weaknesses:**

1. For the ColorMNIST and CelebA, the DEO_M is often comparable to that of the original dataset, showing that the distilled data still may follow the bias of the original dataset, though it hasnt alleviated the bias.
2. One big issue is that it is not clear if the reported scores are statistically signifcant as no std was reported.
3. The only real image dataset for which the analysis was done is CelebA.

**Questions:**

1. The proposed loss function currently considers all groups, but does not consider their cardinality. Would upweighting the minority groups benefit the loss further, where the weights can be inversely proportional to the group size?
2. How robust is this method to the availability of the spurious/group labels? E.g., if a method like JTT [a] is employed to get pseudo labels for the bias attribute, how would the performance change in terms of fairness?
3. What if the original dataset is group balanced first, and then the traditional distillation losses are applied? Would that automatically help reduce the bias?
4. The target attributes reported for CelebA are either attractive, big nose, or young. Attractive and big nose can be subjective. Does the efficacy of the proposed method hold for a more objective attribute like blond hair, which is a famously reported in the fairness literature?


        [a] Liu, Evan Z., et al. "Just train twice: Improving group robustness without training group information." International Conference on Machine Learning. PMLR, 2021.

---

> ### Author Response · Authors · 2024-11-22
> **Response to Reviewer PZx9 (Part 1)**
>
> **W1: For the ColorMNIST and CelebA, the DEO\_M is often comparable to that of the original dataset, showing that the distilled data still may follow the bias of the original dataset, though it hasn't alleviated the bias.**
>
> Thank you for your insightful comments. We attribute the effect of fairness metrics to two primary factors:
>
> 1. `Data: More balanced data generally facilitates model fairness trained on it.`
>
> 2. `Inductive bias of model: The inherent bias of the model itself toward the input data.`
>
> In our work, we focus on generating PA-balanced data. However, PA-balanced data does not necessarily guarantee a fairer model. When the model has an inductive bias toward common patterns shared across PA groups for TA recognition, the importance of PA-balanced data becomes less significant.
>
> Additionally, since our model uses condensed samples compared to the original dataset, this sometimes results in the partial loss of important patterns critical for TA recognition. The extent depends on the specific DD algorithm. Therefore, sometimes, in a certain dataset and metrics, the model trained on the whole dataset may have a good fairness performance.
>
> Combining these two factors, although the models trained on CMNIST and CelebA perform well in terms of fairness, this does not hinder our approach from successfully mitigating the bias present in the original dataset from a data perspective, as demonstrated in Figure 3 of our manuscript.`
>
> **W2: One big issue is that it is not clear if the reported scores are statistically significant as no std was reported.**
>
> Thank you for your feedback. The results in our paper are the averaged values across three runs. In the revised version, we have supplemented the results with the standard deviation, and the final results are presented in the format of mean ± standard deviation in Tables 21-25.
>
> **W3: The only real image dataset for which the analysis was done is CelebA.**
>
> Thank you for pointing out your concerns. We have supplemented another dataset namely UTKFace, commonly used for fairness. It consists of 20,000 face images including three attributes, age, gender, and race. We follow a common setting and treat age as the target attribute and gender as the protected attribute. We test DM and FairDD + DM with the same parameters, the results in Table show that our method outperforms the vanilla dataset distillation by 16.1\% and 8.92\% on the DEO$_M$ and DEO$_A$. Similar results are observed at IPC = 50.
>
> | Methods (Dataset)   | IPC || DM | || DM+FairDD ||
> |----------------------|-----|:------------:|:------------:|:------------:|:------------:|:------------:|:------------:|
> |    | | **Acc.**| **$DEO_M$**| **$DEO_A$**| **Acc.**| **$DEO_M$**| **$DEO_A$**|
> | UTKFace    | 10  | 66.67     |32.97| 16.31 | 67.72| 16.87| 7.39|
> | UTKFace    | 50  | 73.15     |28.58| 14.03 | 74.66| 10.59| 5.09|
>
>
> **Q1: The proposed loss function currently considers all groups, but does not consider their cardinality. Would upweighting the minority groups benefit the loss further, where the weights can be inversely proportional to the group size?**
>
> Thank you for your insightful comments. We denote the model with inversely proportional weighting as FairDD$_ {inverse}$. Our experiments on C-FMNIST and CIFAR10-S at IPC=10 reveal that FairDD$_ {inverse}$ suffers significant performance degradation, with $\text{DEO}_M$ increasing from $33.05\%$ to $56.60\%$ and $\text{DEO}_A$ rising from $19.72\%$ to $35.13\%$ in terms of fairness performance metrics. Additionally, there is also a decline in accuracy for TA.
>
> | Methods (Dataset)   | IPC || DM | |DM| + |  FairDD$_ {inverse}$| | DM+FairDD ||
> |----------------------|-----|:------------:|:------------:|:------------:|:------------:|:------------:|:------------:|:------------:|:------------:|:------------:|
> |    | | **Acc.**| **$DEO_M$**| **$DEO_A$**| **Acc.**| **$DEO_M$**| **$DEO_A$**| **Acc.**| **$DEO_M$**| **$DEO_A$**|
> | C-FMNIST (BG)       | 10  | 22.26     | 100.0| 99.05 | 69.22| 64.25| 41.13| 71.10  | 33.05| 19.72|
> | CIFAR10-S           | 10  | 37.88     | 59.20 | 39.31 |38.14| 48.27| 37.41| 45.17| 31.75| 8.73 |
>
> We attribute this degradation to the excessive penalization of groups with larger sample sizes. The success of FairDD lies in grouping all samples with the same PA into a single group and performing the group-level alignment. Each group contributes equally to the total alignment, inherently mitigating the effects of imbalanced sample sizes across different groups.
>
> However, penalizing groups based on sample cardinality reintroduces an unexpected bias related to group size in the information condensation process. This results in large groups receiving smaller weights during alignment, placing them in a weaker position and causing synthetic samples to deviate excessively from large (majority) groups. Consequently, majority patterns become underrepresented, ultimately hindering overall performance.

---

> ### Author Response · Authors · 2024-11-22
> **Response to Reviewer PZx9 (Part 2)**
>
> **Q2: How robust is this method to the availability of the spurious/group labels? E.g., if a method like JTT [a] is employed to get pseudo labels for the bias attribute, how would the performance change in terms of fairness?**
>
> Thanks for your insightful comment. We agree that evaluating the robustness of spurious group labels could provide more insights. We randomly sample the entire dataset according to a predefined ratio. These samples are randomly assigned to group labels to simulate noise. To ensure a thorough evaluation, we set sample ratios at 10\%, 15\%, 20\%, and 50\%. As shown in the table, when the ratio increases from 10\% to 20\%, the DEO$_M$ results range from 14.93\% to 18.31\% with no significant performance variations observed. These results indicate that FairDD is robust to noisy group labels. However, as the ratio increases further to 50\%, relatively significant performance variations become apparent. It can be understood that under a high noise ratio, the excessive true samples of majority attributes are assigned to minority labels. This causes the minority group center to shift far from its true center and thus be underrepresented.
>
> | Methods (Dataset)   | IPC | | DM | || DM+FairDD |(0%)|| DM+FairDD |(10%)|| DM+FairDD|(15%) || DM+FairDD|(20%)|| DM+FairDD |(50%)||DBSCAN||
> |---------------------------|---|:------------:|:------------:|:------------:|:------------:|:------------:|:------------:|:------------:|:------------:|:------------:|:------------:|:------------:|:------------:|:------------:|:------------:|:------------:|:------------:|:------------:|:------------:|:------------:|:------------:|:------------:|
> |    | | **Acc.**| **$DEO_M$**| **$DEO_A$**| **Acc.**| **$DEO_M$**| **$DEO_A$**| **Acc.**| **$DEO_M$**| **$DEO_A$**| **Acc.**| **$DEO_M$**| **$DEO_A$**| **Acc.**| **$DEO_M$**| **$DEO_A$**| **Acc.**| **$DEO_M$**| **$DEO_A$**| **Acc.**| **$DEO_M$**| **$DEO_A$**|
> | CMNIST (BG)   | 10  | 27.95|100.0|99.11|94.88|13.42|6.77|94.34|16.54|7.81|94.44|17.90|8.61|94.32|18.31|9.20|89.56|66.19|25.97|94.65|14.77|6.94|
>
> JTT [5] is an efficient approach for generating pseudo labels when we do not need to consider the specific attributes a sample belongs to. In other words, JTT only provides a binary label indicating whether the sample is biased according to loss ranking. However, in our study, we focus on fine-grained attribute bias, which requires identifying the specific attributes from multiple attributes for each sample. Consequently, JTT cannot be applied to our research. To provide attribute-level signals, we choose an unsupervised clustering method DBSCAN. Specifically, we do not have any group labels and use DBSCAN to cluster the samples within a batch. The clustering label is regarded as the pseudo group label.
> From Table, FairDD achieves 94.77\% accuracy, and 12.38\% $DEO_M$ and 6.80\% $DEO_A$. This demonstrates the potential of FairDD combined with an unsupervised approach when group labels are unavailable.
>
> **Q3: What if the original dataset is group balanced first, and then the traditional distillation losses are applied? Would that automatically help reduce the bias?**
>
> Thanks for your insightful comments. We synthesized a fair version of CelebA, referred to as CelebA$_ {Fair}$. The target attribute is attractive (attractive and unattractive), and the protected attribute is gender (female and male). In the original dataset, the sample numbers for female-attractive, female-unattractive, male-attractive, and male-unattractive groups are imbalanced. To create a fair version, CelebA$_{Fair}$ samples the number of instances based on the smallest group, ensuring equal representation across all four groups.
> We tested the fairness performance of FairDD and DM at IPC = 10, as well as the performance of models trained on the full dataset. As shown in Table, vanilla DD achieves 14.33\% $DEO_A$ and 8.77\% $DEO_M$. In comparison, the full dataset achieves 3.66\% $DEO_A$ and 2.77\% $DEO_M$. DM still exacerbates bias with a relatively small margin, and this is primarily due to partial information loss introduced during the distillation process. FairDD produces fairer results, achieving 11.11\% $DEO_A$ and 6.68\% $DEO_M$.
>
> | Methods (Dataset)   | IPC || Whole ||| DM || | DM+FairDD ||
> |----------------------|-----|:------------:|:------------:|:------------:|:------------:|:------------:|:------------:|:------------:|:------------:|:------------:|
> |    | | **Acc.**| **$DEO_M$**| **$DEO_A$**| **Acc.**| **$DEO_M$**| **$DEO_A$**| **Acc.**| **$DEO_M$**| **$DEO_A$**|
> | CelebA$_ {Fair}$ | 10  |76.33|3.66|2.77|63.31|14.33|8.77|63.17|11.11|6.68|

---

> > ### Author Response · Authors · 2024-11-22
> > **Response to Reviewer PZx9 (Part 3)**
> >
> > **Q4: Does the efficacy of the proposed method hold for a more objective attribute like blond hair, which is a famously reported in the fairness literature?**
> >
> > | Methods (Dataset)   | IPC || Whole ||| DM || | DM+FairDD ||
> > |----------------------|-----|:------------:|:------------:|:------------:|:------------:|:------------:|:------------:|:------------:|:------------:|:------------:|
> > |    | | **Acc.**| **$DEO_M$**| **$DEO_A$**| **Acc.**| **$DEO_M$**| **$DEO_A$**| **Acc.**| **$DEO_M$**| **$DEO_A$**|
> > | CelebA$^h$      | 10  | 79.44|46.67|26.11|77.66|30.28|20.76|79.71|12.70|8.28|
> >
> > Thank you for your detailed feedback. We have supplemented the experiment by regarding blond hair as the target attribute and gender as the sensitive attribute, resulting in CelebA$^h$. As shown in the Table, FairDD at IPC = 10 achieves the fairness of 12.70\% DEO$_M$ and 8.28\% DEO$_A$, and the accuracy of 77.66\%. FairDD outperforms vanilla DM by 17.58\% and 12.48\% on DEO$_M$ and DEO$_A$. Hence, FairDD consistently outperforms the vanilla DD approach in handling the objective attribute.

---

> > > ### Author Response · Authors · 2024-11-26
> > > **Feedback request**
> > >
> > > The discussion period deadline is approaching, and your feedback is highly valuable to us. If our responses have adequately addressed your concerns, we would sincerely appreciate it if you could consider raising your rating to acknowledge our efforts in addressing your questions.

---

> > > > ### Comment · Reviewer_PZx9 · 2024-11-26
> > > >
> > > > Thanks for the detailed responses. I have gone through all the comments of the authors and the other reviewers. While my concerns are more or less resolved, I agree with Reviewers Yauw and SjG8 that the term "Adversarial Matching" is misleading, and the "pull-and-push" argument that the authors describe does not align with the word adversarial. I maintain my rating, but would strongly suggest revision of this term.

---

> > > > > ### Author Response · Authors · 2024-11-26
> > > > >
> > > > > Thank you for your suggestion. We apologize for any confusion caused by the term "Adversarial Matching". In the revised version, we have replaced it with "Synchronized Matching" (highlighted in orange) for clearer clarification.

---

### Official Review · Reviewer_Yauw · 2024-11-06

**Soundness:** 2
**Presentation:** 3
**Contribution:** 1
**Rating:** 3
**Confidence:** 3

**Summary:**

This paper proposes the idea of fair dataset distillation by ensuring that the protected attribute-based samples provide uniform signals across the groups. This is ensured using a distribution matching objective, uniformly distributed across the protected groups. The authors show results on various benchmarks including synthetic and real-world (CelebA) datasets.

**Strengths:**

- The paper is well-written and clear.
- The experimental results provided are extensive and show improvements.

**Weaknesses:**

- Novelty: The biggest concern for me is the novelty. The idea of class-weighting proposed in this work is classically used in class-imbalanced problems. Further, it has been used in the fairness scenarios as well by multiple works [R1, R2]. Although the results are improved it’s hard to see the ingenuity in the approach, it would be great if the authors could please clarify the differences.


- Adversarial Reference is Vague: The authors mention the formulation to be adversarial, however, the loss doesn’t seem to have an explicit adversarial component. Hence, it would be great to clarify the objective clearly concerning the min-max loss component.


- Theory: Theorem 4.2 is a variant of weighted ERM kind of results, which shows that the weighted loss could be an upper bound to standard ERM kind of loss [R3]. Such weighted results have been extensively studied earlier in literature, hence it’s hard for me to realize the potential of the new results.


- Related Baseline Missing: The following paper [R2] introduces the idea of weighted ERM for fairness with protected sub-groups, using loss weighting and escaping saddle points via SAM [R2, R4] to improve fairness properties. Due to the overlap of the current problem with this, it’s important to compare or contrast the proposed method with a baseline constructed on the basis of these.

[R1] Fairness-aware Class Imbalanced Learning
[R2] Fairness-Aware Class Imbalanced Learning on Multiple Subgroups
[R3] Weighted Empirical Risk Minimization: Sample Selection Bias Correction based on Importance Sampling
[R4] Escaping Saddle Points for Effective Generalization on Class-Imbalanced Data

**Questions:**

Is the weighted centroid matching objective in Eq. 4 correct? It might be insufficient, as two groups of samples with entirely different distributions can still have the same centroid. Could you clarify the distribution matching objective better?

---

> ### Author Response · Authors · 2024-11-22
> **Response to Reviewer Yauw (Part 1)**
>
> Thank you very much for taking the time to review our paper. After proofreading your comments, we think that we must clarify the scope and contribution of our paper:
>
> **Our scope:** Our work explores a new field that bridges fairness and dataset distillation, aiming to mitigate the unfairness of condensed datasets while preserving their accuracy (`condensation fairness`). Importantly, our framework, including DDs in DMF, does not update the model parameters. Instead, it uses randomly initialized neural networks (or trained with few epochs) as non-linear feature transformations
>
> However, your comments primarily discuss fairness in the context of `model fairness`, which refers to training a model that outputs fair logits under class-imbalanced datasets. This approach places emphasis on the model itself and does not consider the process of information condensation. These two concepts—condensation fairness and model fairness—have fundamentally different emphases. Unfortunately, you seem to ignore this and stand on the side of model fairness when evaluating our approach to condensation fairness. Hence, we kindly ask you to reconsider the contribution of our paper, taking into account the distinction between these two types of fairness.
>
> **Our contribution:**
> Our paper is the first to reveal bias inheritance and exacerbation during dataset distillation. To tackle this critical issue, we propose an effective approach that significantly mitigates bias in the condensed datasets.
>
> **Q1: Is the weighted centroid matching objective in Eq. 4 correct? It might be insufficient, as two groups of samples with entirely different distributions can still have the same centroid. Could you clarify the distribution matching objective better?**
>
> Thank you for pointing out your concerns. Eq. 4 is equivalent to Eq. 3 which unified the loss function of DDs in DMF. We rewrite Eq. 3 as Eq. 4 to highlight that the majority groups will dominate the alignment between the original and condensed datasets, leading to bias inheritance in the synthetic dataset.
>
> Centroid matching is widely used in the dataset distillation field. Representative methods such as GM [a] and DM [b] utilize centroid matching to align the distributions between the original and synthetic samples. Especially when the distance metric D is MSE, Eq. 3 is the commonly used distribution alignment approach MMD (Maximum Mean Discrepancy), whose effectiveness has been demonstrated in numerous studies.
>
> Distribution matching treats the embeddings as signals to be aligned. These methods use the same randomly initialized network to extract embeddings from both the original and synthetic datasets, and then employ MMD to measure the distributional discrepancy between them. The corresponding gradients are used to update the synthetic dataset.
>
> > **Reference:**
>
> > [a] Zhao, B., Mopuri, K., & Bilen, H. (2020). Dataset Condensation with Gradient Matching. ArXiv, abs/2006.05929.
>
> > [b] Zhao, B., & Bilen, H. (2021). Dataset Condensation with Distribution Matching. 2023 IEEE/CVF Winter Conference on Applications of Computer Vision (WACV), 6503-6512.
>
>
> **W1: Although the results are improved it’s hard to see the ingenuity in the approach, it would be great if the authors could please clarify the differences.**
>
> Thank you for raising your concerns. We have summarized the following differences between our work and [1] and [2]:
>
> **Different Scopes**
>
> References [1] and [2] focus on improving the classifier fairness trained on class-imbalanced datasets. These methods fall under the field of model fairness, which aims to regularize the model to learn fair representations. However, they do not address information condensation, which is a key objective in our work.
>
> In contrast, our work explores a new research field, condensation fairness, which aims to mitigate the bias of condensed datasets. Condensation fairness encompasses two goals:
>
> 1. Guarantee the information of original datasets to be distilled into the condensed datasets.
>
> 2. During this process, both the bias inherent in the original dataset and the bias exacerbated by vanilla dataset condensation should be mitigated simultaneously.
>
> **Different target**
>
> References [1] and [2] primarily target class-level fairness, addressing class imbalance caused by differing sample sizes across classes, similar to the long-tail learning field.
>
> Our work, on the other hand, mitigates attribute bias instead of class-level bias. Attributes and classes describe the object from different aspects. Our objective is to mitigate attribute bias while preserving class information.
>
> **Different usage**
>
> References [1] and [2] produce tailored fair classifiers, which often have limited applicability to other architectures.
>
> Our method condenses a large dataset into a smaller and fair dataset. Once the fair condensed dataset is obtained, it can be reused to train diverse models across different architectures.

---

> > ### Author Response · Authors · 2024-11-22
> > **Response to Reviewer Yauw (Part 2)**
> >
> > **W2: Adversarial Reference is Vague.**
> >
> > We appreciate your detailed feedback. The reason we refer to our method as adversarial matching is to contrast it with the vanilla DDs used in dataset distillation. Previous work focuses on aligning the synthetic dataset with the original dataset primarily for classification accuracy. In other words, these methods only pull the synthetic dataset toward the single center of the original dataset. In such a case, the majority group dominates the generation of the synthetic dataset, with the minority group being easily neglected.
> >
> > To address this, we propose to simultaneously pull the synthetic dataset toward different groups within each class. Each group attracts the synthetic data toward itself, causing the synthetic data to move farther from other groups. This "pull-and-push" process allows the synthetic dataset to reach a stable equilibrium, preventing it from collapsing into a single group. This is why we refer to our approach as adversarial matching.
> >
> > **W3: Such weighted results have been extensively studied earlier in literature, hence it’s hard for me to realize the potential of the new results.**
> >
> > Thank you for raising your concern. The theorem 4.2 illustrates that our approach could achieve class-level distributional coverage by bounding the vanilla dataset distillation tailored for information condensation. Early studies on ERM (Empirical Risk Minimization) primarily focused on image recognition, but our loss function serves as the distribution matching problem. Furthermore, it seems arbitrary to claim that our results have limited potential simply because our method is based on classical theories. In my view, the most important aspect of evaluating a paper's contribution lies in what it offers to its community and how effectively it inspires researchers, rather than solely on how new theory it builds upon.
> >
> > **W4: Related Baseline Missing**
> >
> > We appreciate you bringing these works to our attention. We will include these references in the Related Work section. The works you mentioned center on training fair classifiers on class-imbalanced datasets, which do not involve information condensation. Our work copes with class-balanced yet imbalanced attributes. It is challenging to transfer these methods to our field directly in a short time. However, in response to your answer, we find a subgroup weight mechanism, namely LDAM$_ {iw}$ proposed in [1]. The mechanism weights each class–group combination based on its smoothed inverse frequency:
> > $\omega_{y,g} = \frac{1 - \beta}{1 - \beta^{N_{y,g}}}$, where $\beta$ is a constant and $N_{y,g}$ is the number of instances belonging to class y and group g.
> > We use the mechanism to weight our different groups called DM+FairDD$_{iw}$. The experiments on C-FMNIST and CIFAR10-S at IPC=10 are as follows:
> >
> > | Methods (Dataset)   | IPC || DM | |DM| + | FairDD$_{iw}$| | DM+FairDD ||
> > |----------------------|-----|:------------:|:------------:|:------------:|:------------:|:------------:|:------------:|:------------:|:------------:|:------------:|
> > |    | | **Acc.**| **$DEO_M$**| **$DEO_A$**| **Acc.**| **$DEO_M$**| **$DEO_A$**| **Acc.**| **$DEO_M$**| **$DEO_A$**|
> > | C-FMNIST (BG)       | 10  | 22.26     | 100.0| 99.05 | 69.22| 64.25| 41.13| 71.10  | 33.05| 19.72|
> > | CIFAR10-S           | 10  | 37.88     | 59.20 | 39.31 | 39.27| 49.88| 36.17| 45.17| 31.75| 8.73 |
> >
> > FairDD$_{iw}$ significantly degrades the fairness performance, with DEO$_M$ dropping from 64.25\% to 33.05\% and from 41.13\% to 19.72\% compared to FairDD.
> >
> > We attribute this degradation to the excessive penalization of groups with larger sample sizes. The
> > success of FairDD lies in grouping all samples with the same PA into a single group and performing
> > the group-level alignment. Each group contributes equally to the total alignment, inherently mitigating
> > the effects of imbalanced sample sizes across different groups.
> >
> > However, penalizing groups based on group number reintroduces an unexpected bias related to group size in the information condensation process. This results in large groups receiving smaller weights during alignment, placing them in a weaker position and causing synthetic samples to deviate excessively from large (majority) groups. Consequently, majority patterns become underrepresented, ultimately hindering overall performance.

---

### Author Response · Authors · 2024-11-22
**General Response**

Dear Reviewers and ACs,

We very much appreciate the insightful and detailed review. We are excited to hear the encouraging comments, particularly that our work `for the first time revealed and addressed the crucial issue of bias inheritance and exacerbation in dataset distillation` that was previously neglected (Reviewer **PZx9** and **j9J1**), and "`inspired the research of fairness dataset synthesis`" (Reviewer **PZx9** and **j9J1**). We are also pleased to hear the positive feedback from all the reviewers, including "`novel approach`" (Reviewer **j9J1**), "`theoretical foundation and comprehensive empirical validation`" (Reviewer **j9J1**, **SjG8**), and "`well-written and easy to follow`" (Reviewer **Yauw** and **SjG8**). While we responded to each of the reviewer comments individually, we also provide a brief summary of the main contents of the rebuttal in response to the reviews:

- In response to the feedback from Reviewer **Yauw** and **SjG8**, we have supplemented the relevant clarification at line 199 in the revised version.

- As recommended by Reviewer **j9J1**, we have supplemented more analysis of computation overhead from input resolution and sample number to provide more practicality discussion.

- To address the concerns raised by Reviewers **Yauw** and **PZx9**, we have added more experiments about the weighting mechanism for different groups.

- We have also addressed the feedback from Reviewers Yauw and PZx9 by incorporating
  important references [1,2,3,4,5] that were previously missing.

We also provide a summary of the main changes made to the revised version of the paper in response to the reviews:

- For Reviewer **PZx9**, we update the reported results with standard deviation in all tables of the main text (`Tables 21-25`). An additional experiment on the common real dataset UTKFace has been added in `Table 11`. Analysis of other target attributes, noise group labels, and balanced original dataset have been supplemented in `Tables 7, 12, and 14`.
- For Reviewer **j9J1**, we supplement the experiment about additional computation overhead in `Table 10`, more protected attributes in `Table 7`, fine-grained group division in `Table 15`, and group underrepresentation in `Table 16`.
- For Reviewer **SjG8**, `Table 17` is added to respond to the exploration of ViT backbone. Exploration of the challenging dataset is presented in `Table 18`. `Figure 13` is supplemented to present the visualization of CelebA.

`We have uploaded the revised paper which includes additional experiments and illustrations to address the feedback from the reviewers. For ease of review, we highlight the revised text in orange. For other questions raised by the reviewers, please see our response to individual questions and concerns below each review.`

> **Reference:**

> [1] Subramanian S, Rahimi A, Baldwin T, et al. Fairness-aware class imbalanced learning[J]. arXiv preprint arXiv:2109.10444, 2021.

> [2] Tarzanagh D A, Hou B, Tong B, et al. Fairness-aware class imbalanced learning on multiple subgroups[C]//Uncertainty in Artificial Intelligence. PMLR, 2023: 2123-2133.

> [3] Vogel R, Achab M, Clémençon S, et al. Weighted empirical risk minimization: Sample selection bias correction based on importance sampling[J]. arXiv preprint arXiv:2002.05145, 2020.

> [4] Rangwani H, Aithal S K, Mishra M. Escaping saddle points for effective generalization on class-imbalanced data[J]. Advances in Neural Information Processing Systems, 2022, 35: 22791-22805.

> [5] Liu, Evan Z., et al. "Just train twice: Improving group robustness without training group information." International Conference on Machine Learning. PMLR, 2021.

---

> ### Author Response · Authors · 2024-11-26
> **Additional General Response**
>
> To address the common confusion about the term "Adversarial Matching", we have replaced it with "Synchronized Matching" (highlighted in orange) in the revised manuscript. We sincerely thank all reviewers for their constructive comments.

---

### Meta-Review · Area_Chair_xpMv · 2024-12-21

**Metareview:**

This submission was reviewed by four reviewers, with discussions involving multiple rounds of clarifications between the reviewers and authors. The opinions were mixed: two reviewers viewed the paper as a borderline case, leaning towards acceptance, while the remaining two reviewers held contrasting accept and reject stances. Overall, the final scores converged near a borderline rating.

While one reviewer provided a high positive score, the review lacked technical depth. The critical reviewer, however, raised significant concerns, including the limited novelty of FairDD compared to existing class-imbalanced learning methods and prior works addressing biases in dataset distillation. Additionally, several reviewers noted that the term "Adversarial Matching" used in the paper is misleading.  Another concern is that the paper assumes the availability of group labels for fair dataset distillation. However, it does not include a baseline comparison where the classifier used in the distillation process is itself trained fairly using established approaches from the literature. This raises an important question: will the distilled dataset  ensure fairness, or will it still exhibit biases? This issue remains unexplored and warrants further experimentation.

After a thorough examination of the reviews and rebuttal, the AC panel concluded that the paper requires significant revisions to address these issues. While it shows potential, the current version does not meet the bar for acceptance. We encourage the authors to incorporate the reviewers' feedback to strengthen the work for future submissions.

**Additional Comments On Reviewer Discussion:**

This paper underwent several rounds of discussion between the reviewers and authors. While the discussions were technical and constructive, most reviewers maintained their original scores.

The authors made an effort to address concerns raised by the critical reviewer regarding the novelty and adversarial objectives of the proposed approach. However, despite these clarifications, the critical reviewer remained unconvinced and retained their negative score.

Given the mixed feedback and unresolved concerns, the AC panel concluded that the paper requires significant revisions before it can be considered for acceptance. We encourage the authors to carefully address the reviewers' comments to strengthen the work for future submissions.

---

### Decision · Program_Chairs · 2025-01-22

Reject